# Gut Helicobacter presentation by multiple dendritic cell subsets enables context-specific regulatory T cell generation

**Emilie V Russler-Germain[1]\*, Jaeu Yi[1], Shannon Young[1], Katherine Nutsch[1], Harikesh S Wong[2], Teresa L Ai[1], Jiani N Chai[1], Vivek Durai[3], Daniel H Kaplan[4], Ronald N Germain[2], Kenneth M Murphy[3], Chyi-Song Hsieh[1]\***

[1]Department of Internal Medicine, Division of Rheumatology, Washington University School of Medicine, St. Louis, United States; [2]Lymphocyte Biology Section, Laboratory of Immune System Biology, National Institute of Allergy and Infectious Diseases, National Institutes of Health, Bethesda, United States; [3]Department of Pathology, Division of Immunobiology, Washington University School of Medicine, St. Louis, United States; [4]Department of Dermatology, Department of Immunology, Pittsburgh Center for Pain Research, University of Pittsburgh, Pittsburgh, United States

**Abstract** Generation of tolerogenic peripheral regulatory T (pTreg) cells is commonly thought to involve CD103$^+$ gut dendritic cells (DCs), yet their role in commensal-reactive pTreg development is unclear. Using two Helicobacter-specific T cell receptor (TCR) transgenic mouse lines, we found that both CD103$^+$ and CD103$^-$ migratory, but not resident, DCs from the colon-draining mesenteric lymph node presented Helicobacter antigens to T cells ex vivo. Loss of most CD103$^+$ migratory DCs in vivo using murine genetic models did not affect the frequency of Helicobacter-specific pTreg cell generation or induce compensatory tolerogenic changes in the remaining CD103$^-$ DCs. By contrast, activation in a Th1-promoting niche in vivo blocked Helicobacter-specific pTreg generation. Thus, these data suggest a model where DC-mediated effector T cell differentiation is 'dominant', necessitating that all DC subsets presenting antigen are permissive for pTreg cell induction to maintain gut tolerance.

**\*For correspondence:**
russler@wustl.edu (EVR-G);
chsieh@wustl.edu (C-SH)

**Competing interests:** The authors declare that no competing interests exist.

## Introduction

The intestinal immune system is required to balance maintaining tolerance to commensal bacteria while preserving the capability of mounting an inflammatory response to pathogenic bacteria. Commensal bacterial induction of tolerogenic peripheral regulatory T (pTreg) cells, and not pro-inflammatory effector T cells, appears to be important for preserving immune homeostasis and health and preventing the development of inflammatory bowel disease (IBD) (*Blander et al., 2017*; *Imam et al., 2018*; *Shale et al., 2013*).

A substantial body of evidence suggests that the process by which naïve CD4$^+$ T cells are selected to be pTreg vs effector T cells is directed by specific conventional dendritic cell (cDC) subsets. For example, cDC1s direct the differentiation of Th1 T cells through their production of IL-12. In a similar fashion, subsets of cDC2s have been shown to be important for both Th2 and Th17 differentiation (*Durai and Murphy, 2016*).

In the intestine, CD103$^+$ cDCs comprising both CD103$^+$ CD11b$^-$ cDC1s and CD103$^+$ CD11b$^+$ cDC2s are commonly thought to induce the differentiation of pTreg cells from naïve T cells (*Hasegawa and Matsumoto, 2018*; *Scott et al., 2011*; *Takenaka and Quintana, 2017*; *Tanoue et al., 2016*). The studies that initially characterized this subset as the predominant pTreg

cell-inducing subset of cDCs were primarily done in vitro and determined that the pTreg cell-inducing properties of mixed CD103$^+$ CD11b$^-$ cDC1 and CD103$^+$ CD11b$^+$ cDC2 populations were dependent on TGF-β, the retinoic acid-producing enzyme retinaldehyde dehydrogenase (RALDH), and the tryptophan-catabolizing enzyme indoleamine 2,3-dioxygenase (IDO) (*Coombes et al., 2007*; *Huang et al., 2013*; *Matteoli et al., 2010*; *Sun et al., 2007*). Importantly, the early studies examining CD103$^+$ cDCs (*Coombes et al., 2007*; *Sun et al., 2007*) were performed before it was appreciated that this population included both cDC1s and a subset of cDC2s. These early studies also did not discriminate between migratory and resident cDC subsets in the mesenteric lymph nodes (MLN) or between CD11c$^+$ cDCs and macrophages in the colon lamina propria (cLP), therefore confounding comparisons of CD103$^+$ and CD103$^-$ cDC subsets in pTreg induction (*Coombes et al., 2007*; *Huang et al., 2013*; *Matteoli et al., 2010*; *Sun et al., 2007*). In addition to these in vitro experiments, in vivo studies have suggested that CD103$^+$ cDC1s and cDC2s are important for ovalbumin (OVA)-specific pTreg cell differentiation in a model of oral tolerance. Esterhazy et al. used Zbtb46$^{Cre}$::Irf8$^{fl/fl}$ mice to deplete cDC1s and Mazzini et al. used Itgax-Cre::Gja1$^{fl/fl}$ mice to delete Connexin 43 in all CD11c$^+$antigen presenting cells (APCs), finding a defect in antigen presentation by small intestine CD103$^+$ CD11b$^+$ cDC2s; however, the conclusion that CD103$^+$ cDC1s and cDC2s are critical may have been confounded by the deletion of these targeted genes in all cDC subsets (*Esterházy et al., 2016*; *Mazzini et al., 2014*). Finally, Welty et al. showed that depletion of CD103$^+$ cDC1s and cDC2s in CD207-DTA::Batf3$^{-/-}$ mice resulted in decreased polyclonal CD4$^+$ T cell numbers in the small intestine lamina propria, including Treg cells (*Welty et al., 2013*). Although this latter study cannot address whether CD103$^+$ cDC1s and cDC2s contributed to pTreg cell induction vs maintenance, these data add to the body of literature supporting the notion that CD103$^+$ cDC1s and cDC2s are important for pTreg cell induction.

Recently, this prevailing concept that CD103$^+$ cDC1s and cDC2s are the primary inducers of pTreg cells in the intestine has been challenged. One study of colonic tolerance to OVA enema showed that protection against delayed-type hypersensitivity (DTH) could be mediated by CD103$^-$ CD11b$^+$ cDC2s (*Veenbergen et al., 2016*). As the draining LN of OVA enema are the iliac and caudal LNs, which naturally lack CD103$^+$ CD11b$^+$ cDC2s, the use of Batf3$^{-/-}$ mice eliminated the only CD103$^+$ cDCs in these LNs, the CD103$^+$ cDC1s. The notion that CD103$^-$ CD11b$^+$ cDC2s are sufficient for inducing OVA-specific Treg cells was supported by in vitro experiments, but in vivo OVA-specific Treg induction after OVA enema in Batf3$^{-/-}$ mice was not assessed. Similarly, we previously found that colonic Helicobacter-specific pTreg cell differentiation was unaffected in Batf3$^{-/-}$ mice (*Nutsch et al., 2016*). However, one caveat of our study is that Batf3-deficiency may only result in a partial decrease in cDC1s in intestinal tissues (*Tussiwand et al., 2012*) and does not eliminate CD103$^+$ CD11b$^+$ cDC2s in the MLN draining more proximal portions of the GI tract. Another study found that MHC Class II expression in CX3CR1$^+$ mononuclear phagocytes, thought to be a subset apart from CD103$^+$ cDC1s and cDC2s (*Bogunovic et al., 2009*), is required for pTreg cell induction in oral tolerance and that CX3CR1$^+$ mononuclear phagocytes are required for pTreg cell differentiation in commensal tolerance in Rag2$^{-/-}$ host mice (*Kim et al., 2018*). These studies therefore argue that non-CD103$^+$ DCs are important or redundant for Treg cell development. However, it is possible that the 'tolerogenic' cDC subset differs for oral or per rectum administered soluble proteins vs. naturally colonized commensal bacterial antigens. Thus, the importance of cDC subsets for the induction of intestinal pTreg cells remains an open area of study (*Mowat, 2018*).

Previously, we identified two colonic T cell clones (CT2 and CT6) that undergo pTreg cell differentiation in response to distinct Helicobacter (H.) species, H. typhlonius or H. apodemus, respectively (*Chai et al., 2017*). Colonic Helicobacter species reside in close proximity to the intestinal epithelium within the mucous layer in the crypts (*Recordati et al., 2009*) and are classified as pathobionts that cause inflammation in susceptible hosts, yet are pervasive in healthy hosts (*Fox et al., 1999*; *Taylor et al., 2007*). At homeostasis, Zbtb46$^+$ cDCs are essential for presenting H. typhlonius and H. apodemus to naïve T cells in the colon-draining distal MLN (dMLN); naïve Helicobacter-specific T cells are not activated in vivo in the absence of cDCs (*Chai et al., 2017*; *Lathrop et al., 2011*; *Nutsch et al., 2016*). However, the specific cDC subset that presents Helicobacter antigens to naïve T cells and mediates pTreg cell selection remains unknown.

Here, we examine the role of cDC subsets in presenting antigens from Helicobacter to naïve T cells and how they influence naïve T cell differentiation. Our data argue against the hypothesis that CD103$^+$ cDC1s and cDC2s represent specialized cDC subsets required for presentation of gut

commensal antigens and pTreg cell induction. Rather, our data support the notion that unlike certain cDC functions, induction of commensal-specific Treg cells in the periphery is not restricted to a specific migratory cDC subset. These data support a model in which pTreg cell development is 'recessive' such that all cDCs presenting cognate antigen to a given naïve T cell must be permissive for the induction of FOXP3, and that the presence of antigen-carrying cDCs that induce canonical effector T cell development 'dominantly' blocks pTreg cell generation.

## Results

### Migratory cDCs present Helicobacter antigens during homeostasis

If a specific subset of cDCs facilitates the conversion of commensal-specific T cells into pTreg cells, then this subset should present Helicobacter antigens on MHC Class II. We therefore sought to determine the cDCs that present Helicobacter antigens in vivo. First, we asked whether Helicobacter antigens are presented by cDCs resident in the dMLN vs those that migrate from the colon. Lymph node resident cDCs may acquire soluble antigens either from lymphatic drainage from the colon or transfer from migratory cDCs (*Allan et al., 2006*; *Hor et al., 2015*; *Sixt et al., 2005*). In contrast, migratory cDCs have been shown to pick up antigens in the intestine lamina propria through a variety of mechanisms and then move through afferent lymphatics to the draining MLN (*Cerovic et al., 2013*; *Farache et al., 2013*; *Mazzini et al., 2014*; *McDole et al., 2012*; *Worbs et al., 2006*).

To directly examine which of these cDC subsets are loaded with Helicobacter antigen in vivo, we sorted resident (MHC II$^{int}$ CD11c$^{hi}$) and migratory (MHC II$^{hi}$ CD11c$^{int}$) cDCs (*Satpathy et al., 2012*) from the dMLN and co-cultured them with naïve Helicobacter-specific T cell receptor (TCR) transgenic cells that recognize *H. typhlonius* (CT2) or *H. apodemus* (CT6) (*Figure 1—figure supplement 1A*; *Chai et al., 2017*). TCR activation was assessed by CD25 upregulation. We observed that migratory cDCs were much more efficient than resident cDCs at activating both CT2 and CT6 TCR transgenic T cells, as assessed by the percentage of activated CD25$^+$ cells and CD25 median fluorescence intensity (MFI), a measurement for magnitude of activation within individual cells (*Figure 1A*). When exogenous autoclaved Helicobacter antigens were added to the cultures, resident cDCs were actually more efficient than migratory cDCs at activating naïve T cells, confirming that resident cDCs are fully capable of antigen uptake, processing, and presentation (*Figure 1B*). Migratory cDC stimulation of T cells was MHC Class II-dependent, as it could be blocked by addition of monoclonal anti-MHC Class II blocking antibody to the culture (*Figure 1—figure supplement 1B*). In summary, these ex vivo data show that migratory, but not resident, cDCs in the dMLN present Helicobacter antigens during homeostasis.

We then addressed whether genetic ablation of a receptor necessary for efficient cDC migration would affect CT2 and CT6 T cell activation in vivo. *Ccr7*$^{-/-}$ mice lack a critical chemokine receptor for cDC trafficking to the lymph node and therefore lack most migratory cDCs in the dMLN (*Mikulski et al., 2015*; *Ohl et al., 2004*; *Satpathy et al., 2012*). One week after transfer of naïve TCR transgenic cells into *Ccr7*$^{+/-}$ or *Ccr7*$^{-/-}$ mice, we observed a striking decrease in T cell activation, as measured by the number of recovered CT2 and CT6 cells in the dMLN after adoptive transfer, as well as a decrease in the induction of Foxp3$^+$ CT2 and CT6 cells (*Figure 1C*). A caveat of these results is that CCR7 is required for the development of normal lymphoid organ architecture and T cell entry into lymph nodes, resulting in altered immune responses in host mice (*Förster et al., 2008*). Nevertheless, these in vivo *Ccr7*$^{-/-}$ results are consistent with the ex vivo cDC results described above, and together these data demonstrate an important role of migratory cDCs in the activation of naïve Helicobacter-specific T cells.

### All migratory cDC subsets present Helicobacter antigens

We next asked which subset(s) of migratory cDCs are capable of activating naïve CT2 and CT6 T cells. For these studies, we sorted single positive CD103$^+$ CD11b$^-$ (CD103$^+$ SP) cDC1, double positive CD103$^+$ CD11b$^+$ (DP) cDC2, and single positive CD103$^-$ CD11b$^+$ (CD11b$^+$ SP) cDC2 subsets of migratory cDCs. We assessed cDC purity using *Zbtb46*$^{GFP}$ mice (*Figure 2—figure supplement 1A*), which express GFP in cDC lineage cells as well as monocyte-derived cDCs during inflammation, but not in macrophages (*Satpathy et al., 2012*). This revealed that about 15–25% of CD103$^-$ CD11b$^+$ sorted cells are F4/80$^{lo/int}$*Zbtb46*$^{GFP-}$ (*Figure 2—figure supplement 1A*) and likely represent a

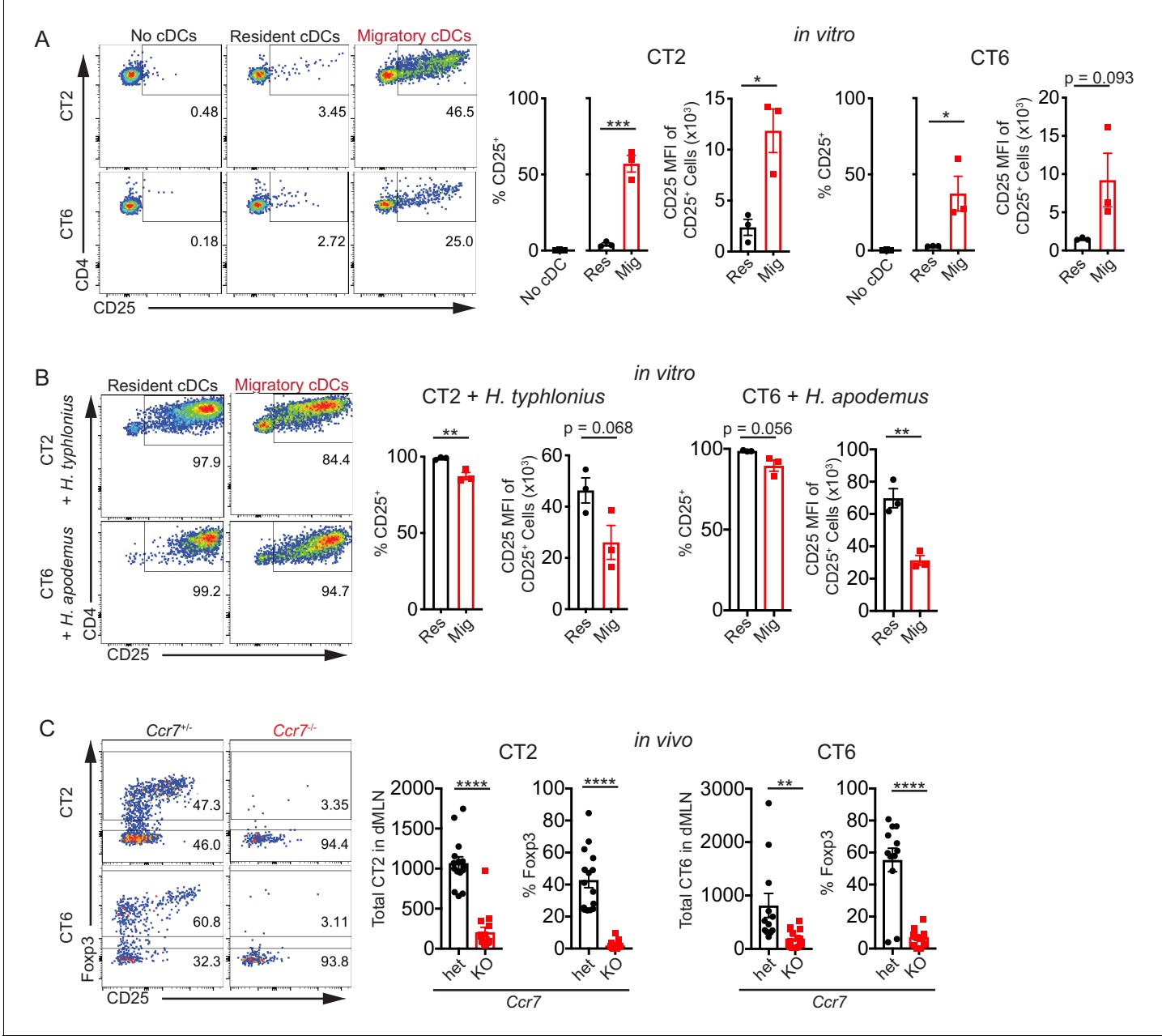

**Figure 1.** Migratory conventional dendritic cells (cDCs) present Helicobacter antigens to naïve T cells. (**A**) Migratory cDCs present endogenously loaded Helicobacter antigens to T cells. Resident (MHCII^int CD11c^hi) or migratory (MHCII^hi CD11c^int) cDCs from the distal mesenteric lymph node (dMLN) were cultured with naïve (CD44^lo CD62L^hi CD4^+) CT2 or CT6 T cells for 2 days. The percentage of CT2 and CT6 T cells that upregulated CD25 and the CD25 median fluorescence intensity (MFI) of CD25^+ cells were analyzed by flow cytometry (expt. = 3). (**B**) Both resident and migratory cDCs can present exogenous Helicobacter antigens. dMLN resident and migratory cDCs were cultured as in (**A**) with the addition of autoclaved isolates of *H. typhlonius* (CT2) or *H. apodemus* (CT6) (expt. = 3). (**C**) Migratory cDCs are necessary for CT2 and CT6 activation in vivo. Naïve CT2 or CT6 (5 × 10^4) were injected into littermate *Ccr7^+/−* or *Ccr7^−/−* mice. Transferred cells in the dMLN were analyzed 1 week later for total cells recovered and *Foxp3*^IRES-GFP or Thy1.1 expression (expt. = 3). Each dot represents an independent experiment except in (**C**), where each dot represents one mouse. Mean ± SEM shown. p-values from Student's t-test (**A–C**) excluding no cDC condition (**A**); *p<0.05, **p<0.01, ***p<0.001, ****p<0.0001. The following figure supplement is available for *Figure 1—figure supplement 1*.

The online version of this article includes the following figure supplement(s) for figure 1:

**Figure supplement 1.** Migratory conventional denditic cells (cDCs) present Helicobacter antigens to naïve T cells.

macrophage subset with cDC-like antigen presentation functionality (*Cerovic et al., 2014*; *Satpathy et al., 2012*). As our previous study using *Zbtb46$^{DTR}$* mice suggested that Zbtb46$^-$ cells played a minor to negligible role in presentation of Helicobacter antigens in vivo (*Nutsch et al., 2016*), we expected that these contaminants would not activate CT2/CT6 in our in vitro analysis of CD103$^-$ CD11b$^+$ cDC2s.

Our previous studies showed a defect in CT2 and CT6 pTreg cell differentiation in Itgax-Cre:: *Notch2$^{fl/fl}$* (Notch2-ΔDC) mice (*Nutsch et al., 2016*). Disruption in NOTCH2 signaling in cDCs results in loss of CD103$^+$ CD11b$^+$ DP cDC2s in the MLN, although development and function of other cDC subsets are also altered in these mice (*Lewis et al., 2011*; *Satpathy et al., 2013*). Based on these previous data and other studies (*Welty et al., 2013*), we predicted that DP cDC2s would be the primary cDC subset presenting Helicobacter antigens in the dMLN. Unexpectedly, we observed that at least some members of all three subsets of migratory cDCs constitutively presented Helicobacter antigens to naïve CT2 and CT6 T cells using the ex vivo culture approach described above (*Figure 2*, *Figure 2—figure supplement 1B*). Although DP cDC2s tended to activate both CT2 and CT6 naïve T cells to a greater degree than the other two subsets, DP cDC2s only constitute about 15% of migratory cDCs in the dMLN and thus may not represent the dominant cDC subset inducing naïve T cell activation in vivo (*Figure 2—figure supplement 1C*). All three subsets of migratory cDCs had comparable naïve T cell activating potential, as demonstrated by coculturing cDCs with naïve OT-II T cells and varying concentrations of OVA peptide (*Figure 2—figure supplement 1D*). Thus, contrary to our hypothesis, these in vitro data show that all migratory cDC subsets carry Helicobacter antigens from the colon and can present the antigens to naïve CT2 and CT6 T cells in the dMLN.

## cDC1s are not required for Helicobacter-specific pTreg cell differentiation in vivo

We next used mice genetically deficient in cDC subsets to study pTreg cell differentiation in vivo. Based on prior studies showing the importance of cDC1s in oral tolerance (*Esterházy et al., 2016*; *Mazzini et al., 2014*), we previously studied cDC1-deficient *Batf3$^{-/-}$* mice (*Nutsch et al., 2016*). We found that CT2 and CT6 pTreg cell development were unchanged in *Batf3$^{-/-}$* mice, consistent with studies examining DTH responses after OVA enema (*Veenbergen et al., 2016*). However, one caveat of *Batf3$^{-/-}$* mice is that a large number of cDC1s remain in mucosal tissue and specifically in the dMLN due to the compensatory expression of *Batf* in a pro-inflammatory environment (*Tussiwand et al., 2012*; *Figure 3—figure supplement 1A*). It was therefore possible that cDC1s in *Batf3$^{-/-}$* mice were sufficient to maintain pTreg cell selection.

In comparison to *Batf3$^{-/-}$* mice, mice which have a 149 bp deletion in the *Irf8* enhancer including the +32 kb BATF3-binding enhancer element (*Irf8$^{\Delta149en/\Delta149en}$*, formerly *Irf8* +32 5'$^{-/-}$*Durai et al., 2019*) showed a much greater reduction of the number of cDC1s in the dMLN (*Figure 3A*). *Irf8$^{\Delta149en/\Delta149en}$* mice did not have a decrease in total migratory cDC numbers, as there were increased numbers of CD103$^+$ CD11b$^+$ and to a lesser extent CD103$^-$ CD11b$^+$ cDC2s in the absence of cDC1s (*Figure 3A*). The frequency of polyclonal colonic CD4$^+$ FOXP3$^+$ T cells was unchanged in *Irf8$^{\Delta149en/\Delta149en}$* mice (*Figure 3B*). Additionally, the proportion of colonic pTreg cells marked by low HELIOS expression (*Thornton et al., 2010*) was unchanged in *Irf8$^{\Delta149en/\Delta149en}$* mice (*Figure 3B*). Although low HELIOS expression is an imperfect marker for pTreg cells, these data suggest that the proportion of bacteria-specific Treg cells is not grossly altered by the loss of CD103$^+$ CD11b$^-$ cDC1s. Finally, both FOXP3$^-$ and FOXP3$^+$ colonic polyclonal T cells exhibited significant decreases in TBET (*Tbx21*) expression, confirming that cDC1s are important for the induction of Th1 and TBET$^+$ Treg cells (*Figure 3B*). Conventional T cells showed increased GATA3 expression, suggesting that compensatory Th2 development occurs with the loss of cDC1s and decrease in Th1 development (*Figure 3B*). RORγt (*Rorc*) expression in colonic conventional T cells remained low after the loss of cDC1s. Together, these data suggest that loss of cDC1s selectively increases colonic Th2 frequencies at the expense of Th1 cells, while polyclonal peripheral FOXP3$^+$ Treg cell proportions are maintained.

To assess the effects of cDC1 loss on commensal antigen-specific pTreg cell development, we transferred naive CT2 and CT6 cells into 3–4-week-old *Irf8$^{\Delta149en/\Delta149en}$* mice. We used 3–4-week-old recipient mice throughout our work to model the natural timeframe of pTreg induction to commensal bacteria as well as food exposure, which occurs at weaning (*Atarashi et al., 2011*; *Nutsch et al., 2016*). Moreover, we previously found that CT2 and CT6 pTreg cell differentiation decreases in mice

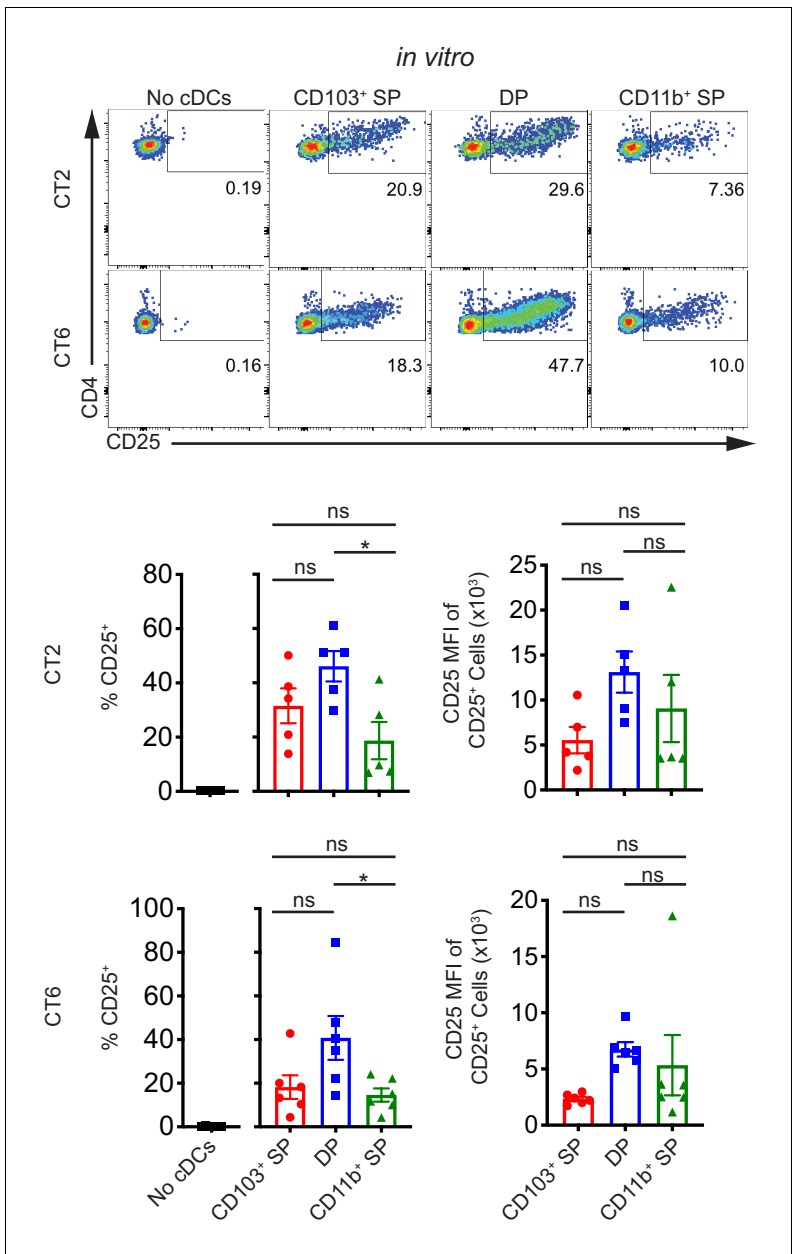

**Figure 2.** Multiple subsets of migratory conventional dendritic cells (cDCs) present Helicobacter antigens to naïve T cells at homeostasis. All three main subsets of migratory cDCs present endogenous Helicobacter antigens to T cells ex vivo. Migratory CD103+ CD11b− cDC1s (CD103+ SP), CD103+ CD11b+ cDC2s (DP), or CD103− CD11b+ cDC2s (CD11b+ SP) were sorted from the distal mesenteric lymph node (dMLN) and cultured with naïve CT2 and CT6 T cells as in *Figure 1A* (expt. = 5–6). Each dot represents an individual experiment. Mean ± SEM is shown. p-values from Tukey's multiple comparisons test excluding no cDC condition; *p<0.05, **p<0.01, ***p<0.001, ****p<0.0001. The following figure supplement is available for *Figure 2—figure supplement 1*.

The online version of this article includes the following source data and figure supplement(s) for figure 2:

**Figure supplement 1.** Multiple subsets of migratory conventional dendritic cells (cDCs)
present Helicobacter antigens to naïve T cells at homeostasis.

**Figure supplement 1—source data 1.** Migratory cDC1s are increased in the dMLN of wild-type mice relative to the proximal MLN (pMLN), *Figure 2—figure supplement 1C* raw data.

**Figure supplement 1—source data 2.** Comparable T cell activation by all subsets of migratory conventional dendritic cells (cDCs), *Figure 2—figure supplement 1D* raw data.

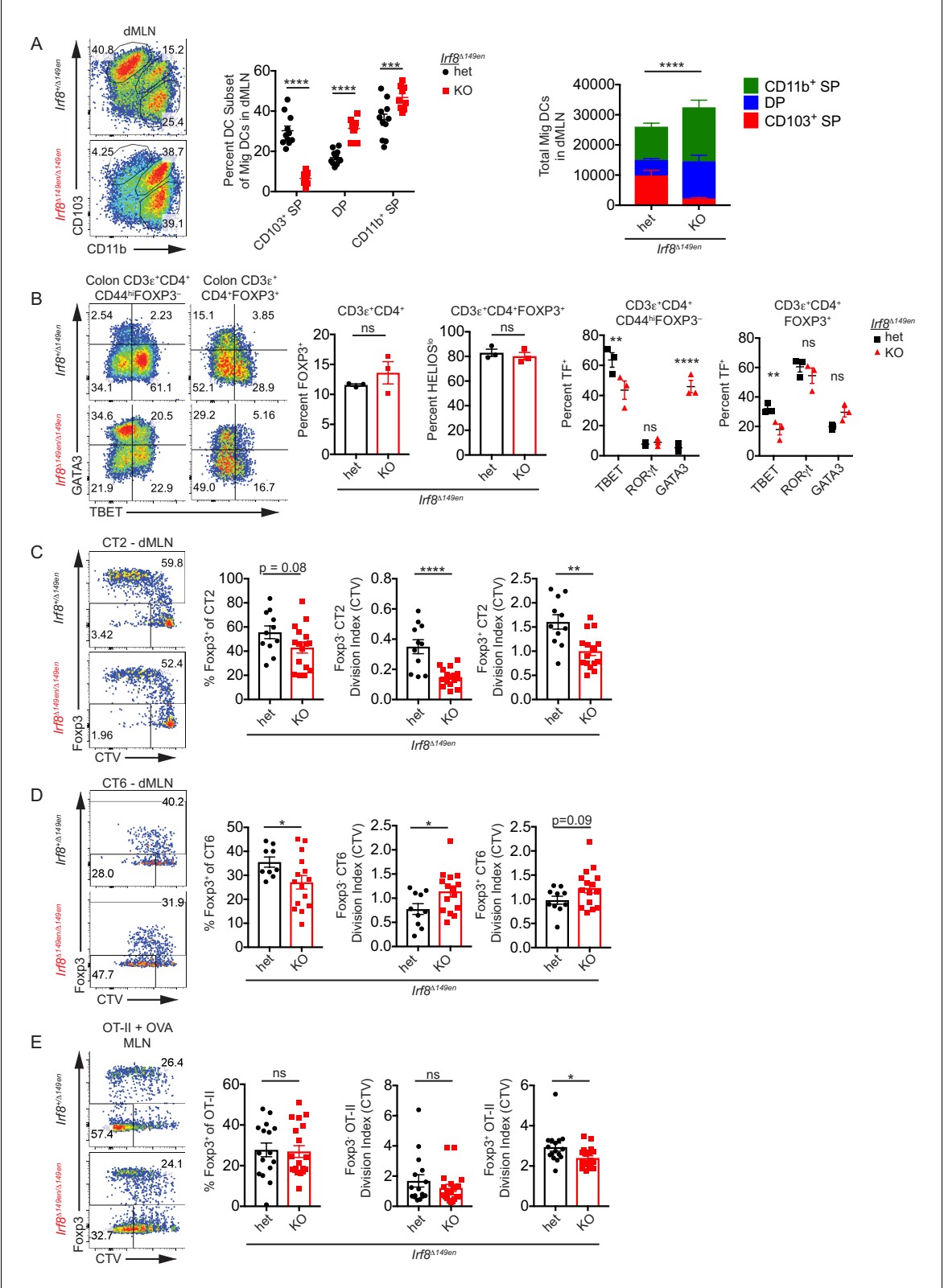

**Figure 3.** CD103+ CD11b− cDC1s are not required for Helicobacter-specific peripheral regulatory T (pTreg) differentiation in vivo. (**A**) Deficiency of CD103+ CD11b− cDC1s in the distal mesenteric lymph node (dMLN) of *Irf8^{Δ149en/Δ149en}* mice. Representative fluorescence-activated cell sorting (FACS) plot and quantification of migratory conventional dendritic cell (cDC) subsets in the dMLN of littermate *Irf8^{+/Δ149en}* and *Irf8^{Δ149en/Δ149en}* mice (expt. = 3, n = 9–11). (**B**) Decreased frequency of Th1 and TBET+ Treg cells in the colon lamina propria (cLP) of *Irf8^{Δ149en/Δ149en}* mice. Representative FACS plots

*Figure 3 continued on next page*

*Figure 3 continued*

and intracellular staining quantification of FOXP3, HELIOS, TBET, RORγt, and GATA3 expression in cLP are shown (expt. = 2). (C and D) Helicobacter-specific T cell activation and Treg cell differentiation are not dramatically altered in cDC1-deficient mice. Congenically marked $5 \times 10^4$ naïve CT2 (C) or $10^5$ naïve CT6 (D) were transferred into littermate $Irf8^{+/\Delta149en}$ and $Irf8^{\Delta149en/\Delta149en}$ mice and analyzed for $Foxp3^{IRES-GFP\ or\ Thy1.1}$ expression and cell trace violet (CTV) dilution in the dMLN after 7 days (expt. = 3 each). (E) T cell activation and Treg cell differentiation in oral tolerance are not altered by loss of cDC1s. $5 \times 10^4$ naïve OT-II cells were transferred into littermate $Irf8^{+/\Delta149en}$ and $Irf8^{\Delta149en/\Delta149en}$ mice, which were gavaged with 50 mg OVA on consecutive days and analyzed as in (C and D) (expt. = 5). Each dot represents an individual mouse. Mean ± SEM (A–E) or + SEM (A, right) shown. p-values from Sidak's multiple comparisons test (A, middle), two-way ANOVA subset/genotype interaction (A, right); Student's t-test (B, middle), Sidak's multiple comparisons test (B, right); Student's t-test (C–E); *p<0.05, **p<0.01, ***p<0.001, ****p<0.0001. The following figure supplement is available for *Figure 3—figure supplement 1*. The following source data are available for (A–E): *Figure 3—source data 1* (A, right).

The online version of this article includes the following source data and figure supplement(s) for figure 3:

**Source data 1.** Deficiency of CD103+ CD11b− cDC1s in the distal mesenteric lymph node (dMLN) of $Irf8^{\Delta149en/\Delta149en}$ mice, **Figure 3A** raw data.

**Figure supplement 1.** CD103+ CD11b− cDC1s are not required for Helicobacter-specific peripheral regulatory T (pTreg) differentiation in vivo.

**Figure supplement 1—source data 1.** Frequency, but not number, of CD103+ cDC1s are reduced in the distal mesenteric lymph node (dMLN) of $Batf3^{-/-}$ mice, **Figure 3—figure supplement 1A** raw data.

greater than 5 weeks of age (*Nutsch et al., 2016*). After such transfer, both CT2 and CT6 exhibited small decreases in pTreg cell differentiation in $Irf8^{\Delta149en/\Delta149en}$ mice, but this was not proportional to the loss of cDC1s (*Figure 3C,D*). The loss of cDC1s had a differential effect on CT2 and CT6 antigen recognition and activation, as CT2 showed decreased, and CT6 showed increased, cell trace violet (CTV) dilution in $Irf8^{\Delta149en/\Delta149en}$ mice (*Figure 3C,D*). Under these conditions, CTV dilution in CT2 and CT6 is entirely dependent on Helicobacter – see Figure 4A in *Chai et al., 2017*. Thus, cDC1s may be more involved in *H. typhlonius* antigen presentation to CT2 than *H. apodemus* presentation to CT6, but they are not essential for Helicobacter-specific pTreg cell differentiation. Furthermore, the selective expansion of DP cDC2s in $Irf8^{\Delta149en/\Delta149en}$ mice and the increase in CTV dilution particularly in Foxp3− CT6 cells may indicate that DP cDC2s are more capable of antigen presentation to naïve CT6 cells than CT2 cells, and that this expansion of DP cDC2s is detrimental to CT6 pTreg cell differentiation.

Since it had previously been reported that Treg cell differentiation was diminished in $Zbtb46^{Cre}::Irf8^{fl/fl}$ mice in a model of oral tolerance (*Esterházy et al., 2016*), we asked whether this was also true for $Irf8^{\Delta149en/\Delta149en}$ mice. Similar to the dMLN, CD103+ SP cDC1s were significantly reduced with a compensatory increase in the CD103+ CD11b+ DP cDC2 population in the proximal MLN (pMLN: all MLN except for dMLN) of $Irf8^{\Delta149en/\Delta149en}$ mice (*Figure 3—figure supplement 1B*). To assess pTreg differentiation in response to oral antigens, we transferred OT-II T cells into OVA-gavaged 3–4-week-old mice to maintain consistency with the CT2/CT6 transfer experiments, which are affected by host age (*Nutsch et al., 2016*), as well as to model oral antigen exposure that takes place at weaning. In contrast to a prior study (*Esterházy et al., 2016*), we did not see a change in the frequency of whole MLN Foxp3+ cells among transferred naïve OT-II cells in $Irf8^{\Delta149en/\Delta149en}$ vs $Irf8^{+/\Delta149en}$ mice fed OVA (*Figure 3E*). We did observe a trend toward decreased T cell activation in $Irf8^{\Delta149en/\Delta149en}$ mice as measured by CTV dilution, similar to what we saw for CT2 (*Figure 3C,E*). The difference between this and the previous study could be due to the use of younger vs older 7–12-week-old hosts (*Esterházy et al., 2016*), or to the nature of the *Irf8* gene locus modification in cDCs between the two mouse lines: $Zbtb46^{Cre}::Irf8^{fl/fl}$ mice lack IRF8 expression in all cDCs (but with incomplete penetrance), while $Irf8^{\Delta149en/\Delta149en}$ mice lack IRF8 expression only in BATF3-dependent cDC1s (*Durai et al., 2019*; *Loschko et al., 2016*). In summary, our data suggest that cDC1s can contribute to the antigen presentation of oral as well as bacterial antigens but are not required for pTreg cell differentiation to those antigens.

## CD103+ CD11b+ cDC2s are not required for Helicobacter-specific pTreg cell differentiation in vivo

We next asked whether the CD103+ CD11b+ (DP) cDC2 subset is required for Helicobacter-specific pTreg cell differentiation, as our in vitro data showed that CT2 and CT6 are most efficiently activated by this subset (*Figure 2*). Others have shown that this subset is preferentially diverted from the lymphatics to the mesenteric fat after *Yersinia pseudotuberculosis* infection, altering the induction of tolerance to commensal microbiota (*Fonseca et al., 2015*). We previously showed that loss of this

subset in Notch2-ΔDC mice results in decreased in vivo CT2 and CT6 pTreg cell differentiation (*Nutsch et al., 2016*). To confirm these results in another model of DP cDC2 deletion, we transferred naïve CT2 and CT6 T cells into Itgax-Cre::*Irf4*$^{fl/fl}$ (Irf4-ΔDC) mice, which exhibit significantly reduced DP cDC2s in the dMLN (*Figure 4—figure supplement 1A*; *Persson et al., 2013*). Similar to Notch2-ΔDC mice, CT2 Foxp3⁻ cells trended toward increased antigen stimulation as measured by CTV dilution (*Figure 4—figure supplement 1B*; *Nutsch et al., 2016*). However, CT2 and CT6 pTreg cell development were unchanged in Irf4-ΔDC mice in contrast to Notch2-ΔDC mice (*Figure 4—figure supplement 1B,C*). This discrepancy suggests that the effects of these genetic modifications on pTreg cell generation were not related to the loss of DP cDC2s, which occurred in both Notch2-ΔDC and Irf4-ΔDC mice, but rather to differences arising from deletion of these genes in the remaining cDCs. We therefore turned to a third genetic model of DP cDC2 depletion that does not require Itgax-Cre mediated gene deletion in all cDCs, but instead uses diphtheria toxin driven by the human langerin promoter (CD207-DTA; formerly huLangerin-DTA) (*Welty et al., 2013*).

We first confirmed that DP cDC2s are missing in the dMLN in CD207-DTA mice, as it was previously only reported for the whole MLN (*Figure 4A*; *Welty et al., 2013*). Since the reduction in cDC2s in CD207-DTA mice can be difficult to appreciate due to spillage of single positive cells into the DP gate, we used additional markers to more specifically quantify DP cDC2s in CD207-DTA mice (*Figure 4—figure supplement 1D*). In the SIRPα⁺ cDC2 population, DP cells were markedly reduced in CD207-DTA mice. Similarly, SIRPα⁺ cDC2s with CD101 expression, which in combination with CD103⁺ is a marker for DP cells (*Bain et al., 2017*), were also significantly reduced in CD207-DTA mice.

Analysis of polyclonal colon T cells in CD207-DTA mice revealed that there was neither a significant increase in Treg cells nor the proportion of HELIOS$^{lo}$ putative pTreg cells (*Figure 4B*). In contrast to previous reports showing that CD103⁺ CD11b⁺ cDC2s are important for Th17 differentiation in the small intestine lamina propria (*Persson et al., 2013*; *Welty et al., 2013*), CD207-DTA mice did not show a decreased percentage of polyclonal Th17 cells or RORγt⁺ Treg cells in the cLP, the former of which was low at homeostasis (*Figure 4B*). Additionally, the expression of other T cell lineage transcription factors in both FOXP3⁻ and FOXP3⁺ T cells were unchanged in colonic polyclonal T cells of CD207-DTA mice (*Figure 4B*). These data suggest that CD103⁺ CD11b⁺ cDC2s, which constitute a small proportion of cDCs within colon-associated dMLN, are not required for colonic T cell differentiation/maintenance at homeostasis.

When naïve CT2 and CT6 T cells were transferred into CD207-DTA mice, they differentiated into Treg cells normally (*Figure 4C,D*). In addition, T cell activation of both Foxp3⁻ and Foxp3⁺ CT2 and CT6 were unchanged in CD207-DTA mice, as measured by CTV dilution (*Figure 4C,D*). Since DP cDC2s constitute a small percentage of cDCs in the dMLN but a much greater percentage in the pMLN, we wondered whether loss of these cells would have a greater effect on Treg cell development in the small-intestine-draining MLN (*Figure 2—figure supplement 1C*). As expected, CD103⁺ CD11b⁺ cDC2s were reduced in the pMLN of CD207-DTA mice, with a compensatory increase in the frequency of CD103⁺ CD11b⁻ cDC1s (*Figure 4—figure supplement 1E*). We therefore examined Treg cell development in oral tolerance after OVA feeding. Surprisingly, given that others have reported a critical role for CD103⁺ CD11b⁺ DP cDCs in oral tolerance Treg cell development (*Mazzini et al., 2014*), OT-II pTreg cell differentiation was significantly increased in CD207-DTA mice without changing T cell activation, indicating that loss of DP cDC2s enhances pTreg cell differentiation in oral tolerance (*Figure 4E*). In summary, these data suggest that DP cDC2s are not required for Helicobacter- or OVA-specific Treg cell generation in the MLN.

## Decreased CD103⁺ cDC1 and cDC2 cell number does not impact Helicobacter-specific pTreg cell differentiation in vivo

At least one study suggested that CD103⁺ cDCs collectively, encompassing both cDC1s and DP cDC2s, are responsible for in vivo Treg cell generation/maintenance in the small intestine (*Welty et al., 2013*). This study utilized CD207-DTA::*Batf3*$^{-/-}$ mice to delete CD103⁺ cDCs. However, as in *Batf3*$^{-/-}$ mice, we have observed that a substantial number of cDC1s can sometimes remain in CD207-DTA::*Batf3*$^{-/-}$ mice (*Figure 5—figure supplement 1A*). When naïve CT2 and CT6 T cells were transferred into CD207-DTA::*Batf3*$^{-/-}$ mice, they showed no decrease in pTreg cell differentiation (*Figure 5—figure supplement 1B,C*). Both CT2 and CT6 showed decreased T cell activation as measured by CTV dilution (*Figure 5—figure supplement 1B,C*). Thus, partial deletion of CD103⁺

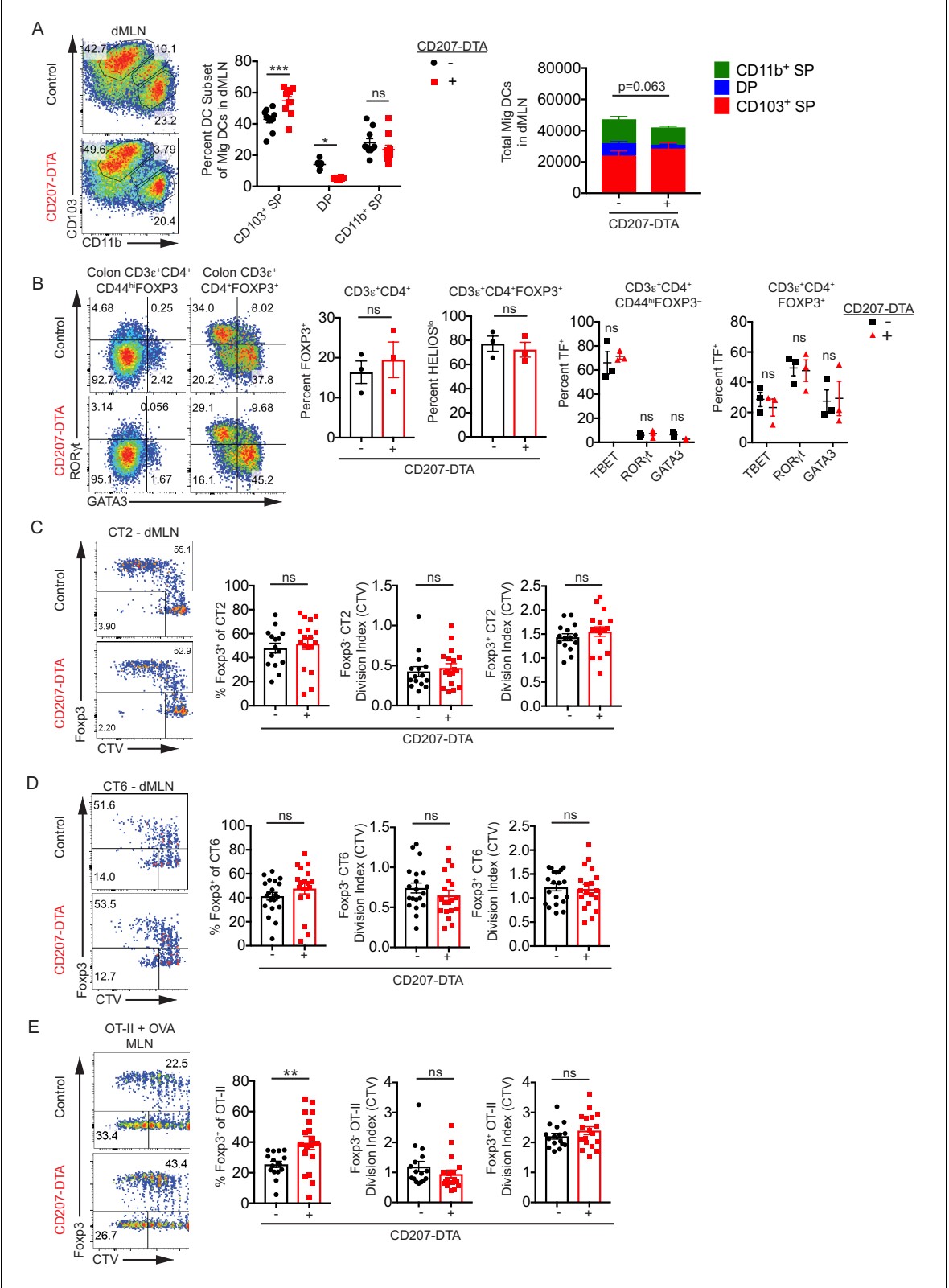

**Figure 4.** CD103⁺ CD11b⁺ cDC2s are not necessary for in vivo Helicobacter-specific peripheral regulatory T (pTreg) differentiation. (**A**) Double positive CD103⁺ CD11b⁺ (DP) cDC2s are lost in the distal mesenteric lymph node (dMLN) of CD207-DTA mice. Representative fluorescence-activated cell sorting (FACS) plot and quantification of migratory conventional dendritic cell (cDC) subsets in the dMLN of littermate control (WT, *Batf3*⁺/⁻, or *Irf8*⁺/Δ149en mice) and CD207-DTA mice (expt. = 7, n = 10). (**B**) The frequencies of T cell subsets in the colon lamina propria (cLP) are unchanged in CD207-

*Figure 4 continued on next page*

Figure 4 continued

DTA mice. Representative FACS plots and intracellular staining quantification of FOXP3, HELIOS, TBET, RORγt, and GATA3 expression in cLP are shown (expt. = 3). (C and D) Helicobacter-specific T cell activation and Treg cell differentiation are not altered in DP cDC2-deficient mice. Congenically-marked $5 \times 10^4$ naïve CT2 (C) or $10^5$ naïve CT6 (D) were transferred into littermate control and CD207-DTA mice and analyzed for Foxp3$^{\text{IRES-GFP or Thy1.1}}$ expression and cell trace violet (CTV) dilution in the dMLN after 7 days (expt. = 8, 7, respectively). (E) Treg cell differentiation in oral tolerance is increased with the loss of DP cDC2s. $5 \times 10^4$ naïve OT-II cells were transferred into littermate WT and CD207-DTA mice, which were gavaged with 50 mg OVA on consecutive days and analyzed as in (C and D) (expt. = 5). Each dot represents an individual mouse. Mean ± SEM (A–E) or + SEM (A, right). p-values from Sidak's multiple comparisons test (A, middle), two-way ANOVA subset/genotype interaction (A, right); Student's t-test (B, middle), Sidak's multiple comparisons test (B, right); Student's t-test (C–E); *p<0.05, **p<0.01, ***p<0.001, ****p<0.0001. The following figure supplement is available for Figure 4—figure supplement 1. The following source data are available for (A–E): Figure 4—source data 1 (A, right).

The online version of this article includes the following source data and figure supplement(s) for figure 4:

**Source data 1.** Double positive CD103$^+$ CD11b$^+$ (DP) cDC2s are lost in the distal mesenteric lymph node (dMLN) of CD207-DTA mice, Figure 4A raw data.

**Figure supplement 1.** CD103$^+$ CD11b$^+$ cDC2s are not necessary for in vivo Helicobacter-specific peripheral regulatory T (pTreg) differentiation.

cDCs resulted in decreased Helicobacter antigen presentation without affecting pTreg cell differentiation.

To more efficiently delete CD103$^+$ cDCs in the dMLN, we bred Irf8$^{\Delta149en/\Delta149en}$ mice to CD207-DTA mice. Unlike straight Irf8$^{\Delta149en/\Delta149en}$ mice, CD207-DTA::Irf8$^{\Delta149en/\Delta149en}$ mice did not have a large increase in DP cDC2s when cDC1 generation was blocked (Figure 5A). Some CD103$^+$ cDC1s and cDC2s remained in the dMLN of these mice, presumably due to compensatory cDC development/survival when a large fraction of cDCs are deleted (Figure 5A). Remaining cDC1s expressed canonical cDC1 makers such as XCR1 and CD36 and thus seem to be bona-fide cDC1s (Figure 5—figure supplement 1D). cLP polyclonal Treg cell frequencies as well as the proportion of putative commensal-specific HELIOS$^{lo}$ Treg cells were unchanged in CD207-DTA::Irf8$^{\Delta149en/\Delta149en}$ mice, indicating that substantial loss of CD103$^+$ cDC1s and cDC2s does not broadly affect Treg cell development (Figure 5B). In contrast to Irf8$^{\Delta149en/\Delta149en}$ mice (Figure 3B), CD207-DTA::Irf8$^{\Delta149en/\Delta149en}$ mice did not show significant decreases in Th1 and TBET$^+$ Treg cells and only a minor, non-significant, increase in colonic GATA3$^+$ Th2 cells (Figure 5B), suggesting that effects of cDC1-deficiency on Teff cell generation may be offset by the loss of DP cDC2s.

When we transferred CT2 and CT6 into CD207-DTA::Irf8$^{\Delta149en/\Delta149en}$ mice to assess the antigen-specific contribution of CD103$^+$ cDC1s and cDC2s to pTreg cell development, we observed no decreases in Helicobacter-specific pTreg cells (Figure 5C,D). Both Foxp3$^-$ CT2 and CT6 showed decreased cell activation by CTV dilution in CD207-DTA::Irf8$^{\Delta149en/\Delta149en}$ mice, in contrast to the increased cell activation seen in Foxp3$^-$ CT6 cells in Irf8$^{\Delta149en/\Delta149en}$ mice (Figure 3D, Figure 5C,D). To determine if pTreg cell differentiation of CT2 and CT6 in CD207-DTA::Irf8$^{\Delta149en/\Delta149en}$ mice was conserved after trafficking to peripheral tissues, we analyzed CT2 and CT6 cells recovered from the cLP 7 days after naïve T cell transfer. Similar to the dMLN, CT2 and CT6 pTreg cell differentiation were not decreased while Foxp3$^-$ cells showed trends of decreased antigen stimulation based on CTV dilution (Figure 5—figure supplement 2A,B). As CD103$^+$ cDC1s and cDC2s in CD207-DTA::Irf8$^{\Delta149en/\Delta149en}$ mice were not completely absent, we wondered whether CT2 and CT6 pTreg cell differentiation correlated with the loss of cDC subsets in the dMLN. Notably, the loss of CD103$^+$ cDCs (cDC1 and/or DP) did not significantly correlate with decreases in pTreg cell induction (Figure 5E, Figure 5—figure supplement 2C). Rather, we saw an increase in the percent of Foxp3$^+$ CT6 cells with decreased CD103$^+$ DC1s, which trended similarly for CT2. Thus, decreased antigen presentation capability with decreased CD103$^+$ cDC1s and cDC2s in vivo did not significantly impact Helicobacter-specific pTreg cell differentiation. However, the maintenance of some CD103$^+$ cDC1s and cDC2 in CD207-DTA::Irf8$^{\Delta149en/\Delta149en}$ mice represents a caveat to the interpretation of the in vivo data presented here.

## Loss of cDC subsets does not markedly alter the colon bacterial microbiome

Perturbations of the microbiota due to cDC subset loss could affect antigen-specific pTreg cell differentiation (Brown et al., 2019). We therefore performed 16S rRNA sequencing and amplicon sequence variant (ASV) analysis of colonic bacteria. Shannon alpha diversity was unchanged in the

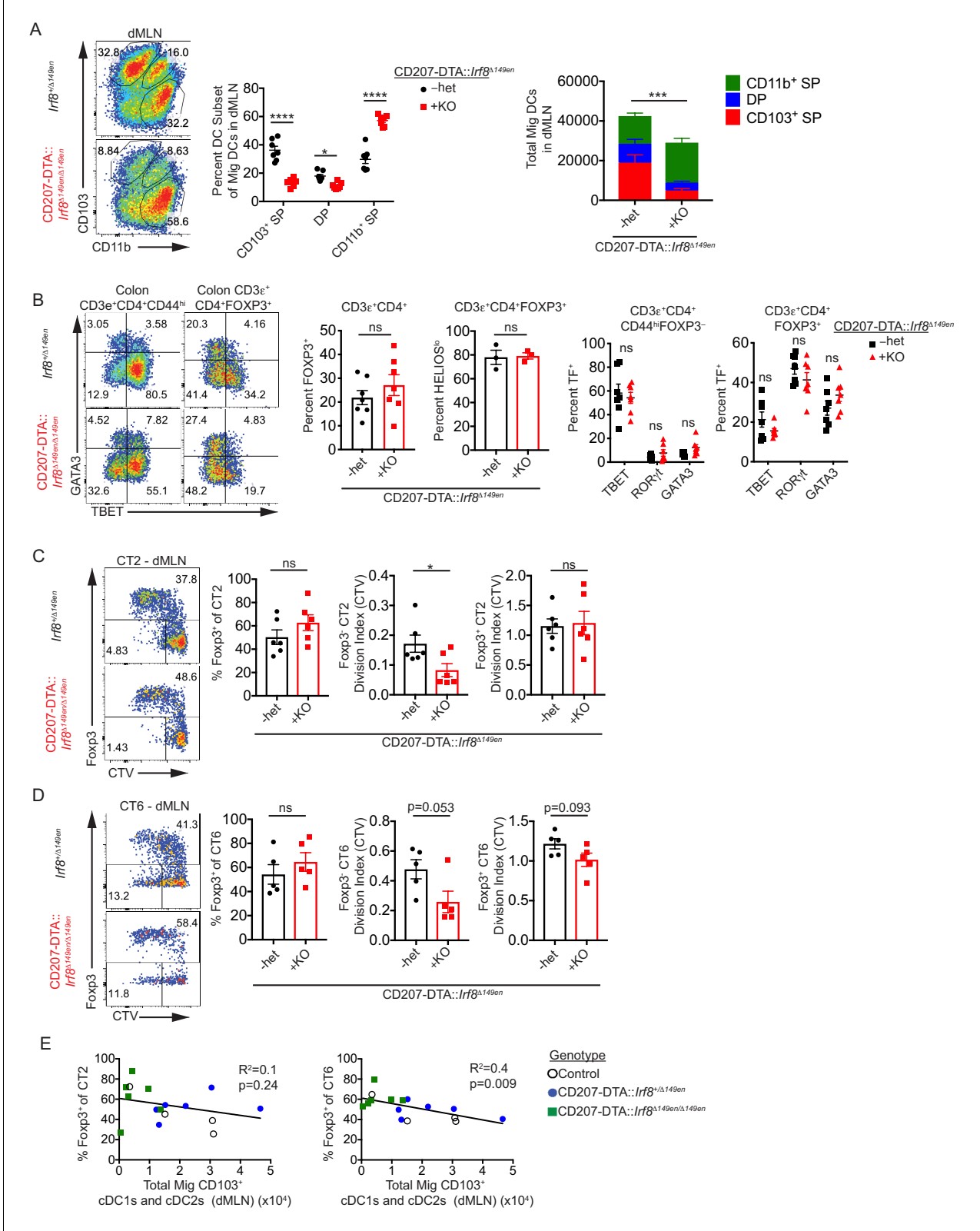

**Figure 5.** CD103⁻ CD11b⁺ cDC2s are sufficient for in vivo Helicobacter-specific peripheral regulatory T (pTreg) differentiation. (**A**) CD103⁺ cDC1s and cDC2s are greatly reduced in the distal mesenteric lymph node (dMLN) of CD207-DTA::*Irf8^Δ149en/Δ149en* mice. Representative fluorescence-activated cell sorting (FACS) plot and quantification of migratory conventional dendritic cell (cDC) subsets in the dMLN of littermate *Irf8^+/Δ149en* and CD207-DTA:: *Irf8^Δ149en/Δ149en* mice (expt. = 7, n = 7). (**B**) The frequency of polyclonal pTreg cells is unchanged in CD207-DTA::*Irf8^Δ149en/Δ149en* mice, but Th1 and

*Figure 5 continued on next page*

Figure 5 continued

TBET$^+$ Treg cells are decreased. Representative FACS plots and intracellular staining quantification of FOXP3, HELIOS, TBET, RORγt, and GATA3 expression in colon lamina propria (expt. = 3–4). (**C and D**) Helicobacter-specific T cell activation and Treg cell differentiation are not dramatically altered in CD103$^+$ cDC1 and cDC2-deficient mice. Congenically marked 5 × 10$^4$ naïve CT2 (**C**) or 10$^5$ naïve CT6 (**D**) were transferred into littermate $Irf8^{+/\Delta149en}$ and CD207-DTA::$Irf8^{\Delta149en/\Delta149en}$ mice and analyzed for $Foxp3^{IRES-GFP\ or\ Thy1.1}$ expression and cell trace violet (CTV) dilution in the dMLN after 7 days (expt. = 4 for both). (**E**) CT6 pTreg cell differentiation is inversely correlated with the number of CD103$^+$ cDC1s and cDC2s in the dMLN. Congenically marked 5 × 10$^4$ naïve CT2 (left) or 10$^5$ naïve CT6 (right) were transferred into littermate control, CD207-DTA, and CD207-DTA::$Irf8^{\Delta149en/\Delta149en}$ mice and analyzed after 7 days. CT2 and CT6 $Foxp3^{IRES-GFP\ or\ Thy1.1}$ expression was compared to cDC subset numbers in the same dMLNs (expt. = 2). Each dot represents an individual mouse. Mean ± SEM (**A–D**) or + SEM (**A**, right). p-values from Sidak's multiple comparisons test (**A**, middle), two-way ANOVA subset/genotype interaction (**A**, right); Student's t-test (**B**, middle), Sidak's multiple comparisons test (**B**, right); Student's t-test (**C and D**); R$^2$ and p-value for nonzero slope (**E**); *p<0.05, **p<0.01, ***p<0.001, ****p<0.0001. The following figure supplements are available for *Figure 5—figure supplement 1* and *Figure 5—figure supplement 2*.

The online version of this article includes the following source data and figure supplement(s) for figure 5:

**Source data 1.** CD103$^+$ cDC1s and cDC2s are greatly reduced in the distal mesenteric lymph node (dMLN) of CD207-DTA::$Irf8^{\Delta149en/\Delta149en}$ mice, *Figure 5A* raw data.

**Figure supplement 1.** CD103$^-$ CD11b$^+$ cDC2s are sufficient for in vivoHelicobacter-specific peripheral regulatory T (pTreg) differentiation.

**Figure supplement 1—source data 1.** Variable frequencies of CD103$^+$ conventional dendritic cells (cDCs) in the distal mesenteric lymph node (dMLN) of CD207-DTA::$Batf3^{-/-}$mice, *Figure 5—figure supplement 1A* raw data.

**Figure supplement 2.** CD103$^-$ CD11b$^+$ cDC2s are sufficient for in vivo Helicobacter-specific peripheral regulatory T (pTreg) differentiation.

cDC-subset deficient mice (*Figure 6—figure supplement 1A*). Additionally, NMDS analyses of bacterial community compositions by Bray–Curtis or UniFrac distances did not show obvious clustering by genotype (*Figure 6—figure supplement 1B*), consistent with a nonsignificant PERMANOVA analysis. Finally, the frequencies of *H. typhlonius* and *H. apodemus* ASVs were not markedly changed by cDC subset deficiency (*Figure 6—figure supplement 1C*). While additional samples would likely improve power to detect differences in the microbiome due to cDC-subset deficiency, our data do not support the hypothesis that microbial changes affected Helicobacter-specific pTreg cell differentiation.

## Loss of CD103$^+$ cDC1s and cDC2s does not increase the tolerogenic potential of migratory CD103$^-$ CD11b$^+$ cDC2s in vivo

Our data collectively suggest that pTreg cell induction is not confined to a single cDC subset and that migratory CD103$^-$ CD11b$^+$ cDC2s are likely sufficient for Helicobacter-specific pTreg cell induction in vivo. However, it remained possible that after genetic deletion of CD103$^+$ cDC1s and cDC2s, the remaining CD103$^-$ CD11b$^+$ cDC2s gain access to niches that facilitate their development into 'tolerogenic' cDCs. To assess this possibility, we quantified the expression of proteins and genes reported to be Treg cell-inducing (*Coombes et al., 2007*; *Hasegawa and Matsumoto, 2018*; *Marie et al., 2005*; *Morris et al., 2003*) in CD103$^-$ CD11b$^+$ cDC2s from CD207-DTA::$Irf8^{\Delta149en/\Delta149en}$ mice. T cell activation-associated proteins (MHC Class II (IAb), CD80, CD86, CD40, PD-L1, and PD-L2) were unchanged in CD103$^-$ CD11b$^+$ cDC2s from CD207-DTA::$Irf8^{\Delta149en/\ \Delta149en}$ mice (*Figure 6—figure supplement 2A*). Additionally, the activity of RALDHs, of which RALDH2 in CD103$^+$ cDC1s and cDC2s has been shown to enhance pTreg cell generation in vitro (*Coombes et al., 2007*; *Sun et al., 2007*), was unchanged in CD103$^-$ CD11b$^+$ cDC2s from CD207-DTA::$Irf8^{\Delta149en/\Delta149en}$ mice (*Figure 6A*). Finally, genes (*Tgfβ1–3*, *Itgb6*, *Itgb8*) and proteins (TGFβ1, CD51) associated with TGF-β production and activation, a molecule that is critical for pTreg cell FOXP3 induction (*Chen et al., 2003*), were not increased and for some genes (*Tgfb2* and *Tgfb3*) were decreased in CD103$^-$ CD11b$^+$ cDC2s from CD207-DTA::$Irf8^{\Delta149en/\Delta149en}$ mice (*Figure 6B,C*). The integrins αvβ6 and αvβ8 activate TGF-β, and αvβ8 expression in cDC1s has been specifically associated with tolerance in the intestines (*Boucard-Jourdin et al., 2016*; *Morris et al., 2003*; *Travis et al., 2007*). While our data confirm previous studies showing preferential function of RALDH and expression of *Itgb8* in cDC1s (*Boucard-Jourdin et al., 2016*; *Coombes et al., 2007*; *Sun et al., 2007*) and *Ido1* in CD103$^+$ CD11b$^+$ cDC2s (*Matteoli et al., 2010*), we found that other Treg cell inducing- and tolerance-associated genes and proteins such as CD51 (*Itgav*), *Tgfb1*, and *Il10* were preferentially expressed in CD103$^-$ CD11b$^+$ cDC2s. However, the increased expression of *Il10*, *Itgav*, and *Tgfb1* may be due to the contamination of the CD103$^-$ CD11b$^+$ cDC2 subset by macrophages (*Figure 2—figure*

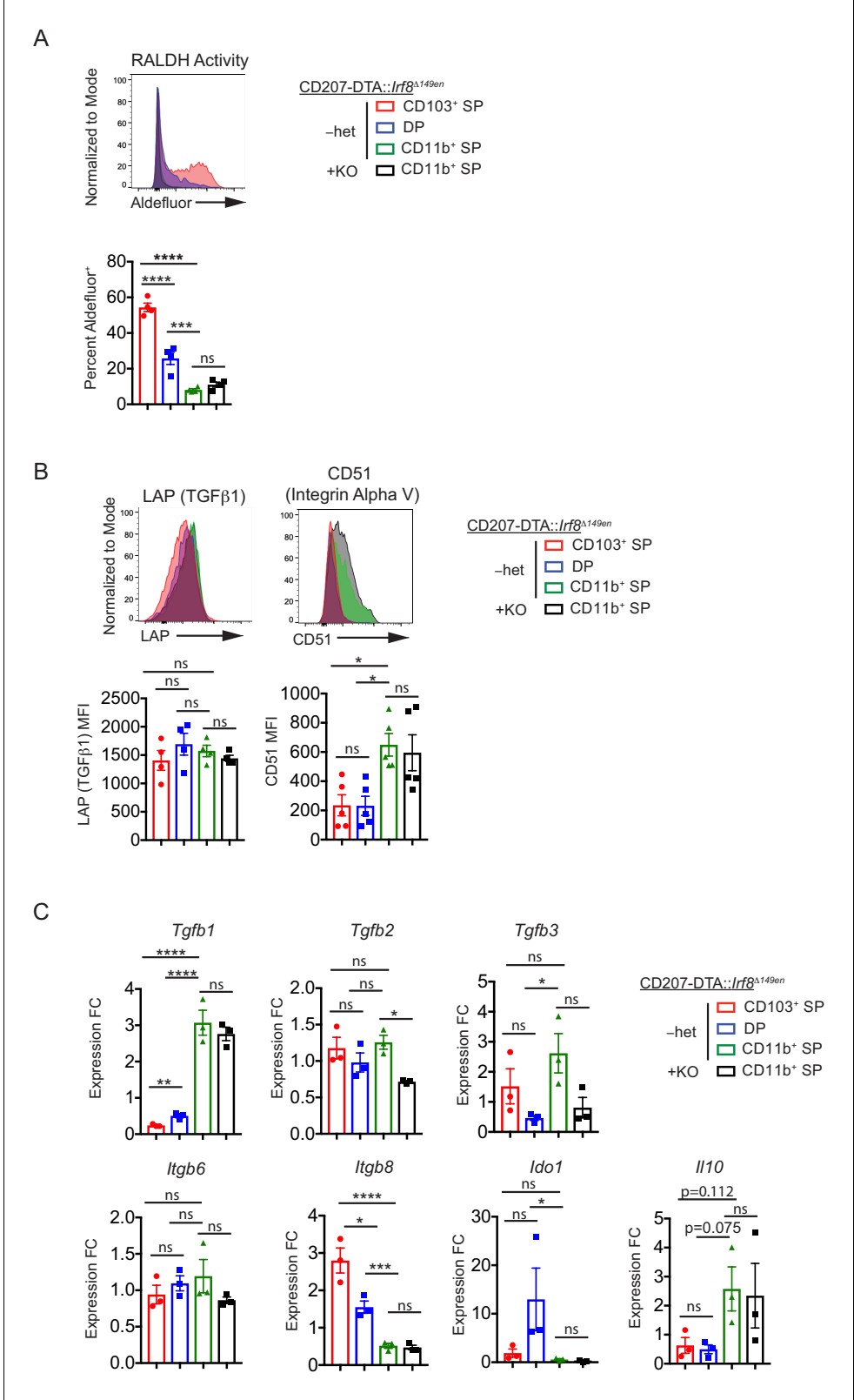

**Figure 6.** CD11b⁺ SP cDC2s are tolerogenic by 'nature', not 'nurture'. (**A**) Retinal dehydrogenase (RALDH) activity as measured by Aldefluor is unchanged in CD11b⁺ SP cDC2s in CD207-DTA::*Irf8*^Δ149en/Δ149en mice deficient in CD103⁺ cDC1s and cDC2s (expt. = 2). (**B**) LAP and CD51 protein expression is unchanged in CD11b⁺ SP cDC2s in CD207-DTA::*Irf8*^Δ149en/Δ149en mice (expt. = 2). (**C**) *Tgfb1*, *IL10*, *Itgb6*, *Itgb8*, and *Ido1* gene expression are unchanged in CD11b⁺ SP cDC2s in CD207-DTA::*Irf8*^Δ149en/Δ149en mice, while *Tgfb2* and *Tgfb3* gene expression are decreased (expt. = 2). Each dot

*Figure 6 continued on next page*

*Figure 6 continued*

represents an individual mouse. Mean ± SEM shown (**A–C**). p-values from Tukey's multiple comparisons test (**A–C**) using Δ-ΔCt values in (**C**); *p<0.05, **p<0.01, ***p<0.001, ****p<0.0001.

The online version of this article includes the following source data and figure supplement(s) for figure 6:

**Figure supplement 1.** CD11b$^+$ SP cDC2s are tolerogenic by 'nature', not 'nurture'.

**Figure supplement 1—source data 1.** 16S rRNA sequencing of whole colon lumen feces of conventional dendritic cell (cDC)-deficient mice, *Figure 6—figure supplement 1B* raw data.

**Figure supplement 2.** CD11b$^+$ SP cDC2s are tolerogenic by 'nature', not 'nurture'.

*supplement 1A*). These data therefore support the idea that each subset of cDCs may employ different mechanisms to induce pTreg cell development (*Figure 6A–C*, *Figure 6—figure supplement 2A*). We also analyzed the remaining CD103$^+$ cDC1s and cDC2s in CD207-DTA::*Irf8$^{\Delta149en/\Delta149en}$* mice to determine if their expression of tolerogenic markers was altered. Remaining CD103$^+$ cDC1s and cDC2s showed significantly decreased CD86 expression and an increased trend of CD274 (PD-L1) expression, suggesting that they may be more 'immature' compared to their wild-type counterparts (*Figure 6—figure supplement 2B*). Taken together, these data suggest that pTreg cell selection in the context of CD103$^+$ cDC deletion is not due to 'nurture' of CD103$^-$ cDCs to become tolerogenic but suggests that the 'nature' of CD103$^-$ CD11b$^+$ cDC2s in normal mice is to induce Helicobacter-specific pTreg cell differentiation.

## Antigen dose-dependent superiority of CD103$^+$ cDC1s for in vitro pTreg cell selection

Our in vivo data are in marked contrast with previous in vitro studies (*Coombes et al., 2007*; *Sun et al., 2007*) regarding the importance of CD103$^+$ DCs, which in their study included both cDC1s and cDC2s. In vitro, DP cDC2s presenting Helicobacter antigens acquired in vivo were the most efficient at inducing Foxp3 in CT2 and CT6 T cells (*Figure 7—figure supplement 1A*). However, the frequency of Foxp3$^+$ cells was much lower than that seen in the in vivo T cell transfer studies (*Figures 3–5*), The variance in Foxp3 induction ex vivo may be attributed to the level of antigen loaded in vivo, as Foxp3 upregulation was correlated with the extent of T cell stimulation by each cDC subset as assessed by the frequency of CD25$^+$ cells (*Figure 7—figure supplement 1B*).

To experimentally control the level of antigen presentation, we stimulated naïve OT-II T cells in vitro with varying concentrations of OVA peptide presented by sorted cDC subsets (*Figure 2—figure supplement 1D*, *Figure 7A,B*). Consistent with previous reports on the total CD103+ cDC population (*Coombes et al., 2007*; *Sun et al., 2007*), we found that CD103$^+$ SP cDC1s were the most efficient at inducing Foxp3 in naïve OT-II T cells in vitro (*Figure 7A*). However, induction of Foxp3 varied according to TCR signal strength. At lower levels of TCR-stimulation with OVA peptide, all subsets of cDCs were able to induce Foxp3 equivalently, whereas CD103$^+$ CD11b$^-$ cDC1s were more efficient at higher levels of antigen stimulation (*Figure 7A*). This was also true if CD25 was used as a marker for T cell activation to control for possible differences in costimulation and MHC Class II levels between cDC subsets (*Figure 7B*). Whether these similarities or differences between cDC subsets observed in vitro reflect their ability to induce Foxp3 in vivo remains unclear. However, this in vitro assay does not appear to inform on pTreg cell selection to Helicobacter in vivo, perhaps due to uneven levels of antigen acquisition by each subset, reinforcing the need for in vivo studies of cDC subsets and pTreg cell selection.

## A recessive model of pTreg cell selection

Our data do not support a model in which a single cDC subset is critical for inducing pTreg cell selection, and rather suggest that multiple cDC subsets are involved. However, the ability of multiple cDC subsets to induce FOXP3 may be problematic for responses to infection or injury. Evidence suggests that cDC subsets may have unique capacities to induce effector T cell subsets (*Durai and Murphy, 2016*), supported by their different cytokine producing functions but also by their divergent abilities to sense specific bacterial components; for example, MLN CD103$^+$ CD11b$^-$ cDC1s have relatively high expression of *Tlr9* and CD103$^+$ CD11b$^+$ cDC2s have relatively high expression of *Tlr4*, *Tlr6*, and *Nod2* (*Esterházy et al., 2016*). CD103$^+$ CD11b$^+$ cDC2s also express TLR5, which senses

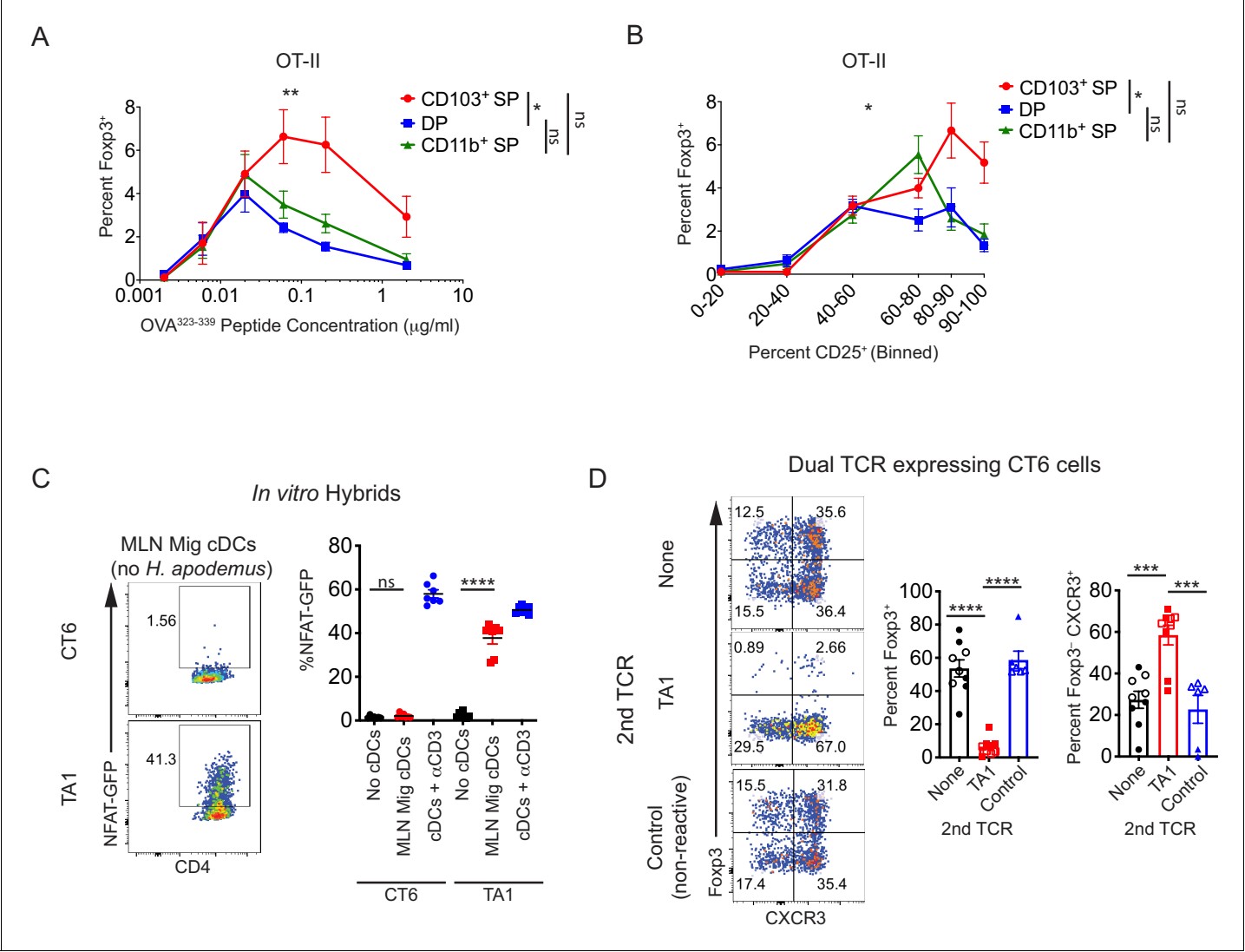

**Figure 7.** Conventional dendritic cell (cDC)-mediated peripheral regulatory T (pTreg) cell differentiation in vivo is recessive to effector T cell development. (A) Migratory cDC induction of OT-II pTreg cells in vitro is dependent on T cell receptor (TCR) stimulation. Treg cell induction of OT-II cells by migratory cDC subsets with varying concentrations of OVA$^{323-339}$ peptide is shown (expt. = 5). (B) In vitro, CD103$^+$ SP cDC1s have increased ability to induce OT-II pTreg cells at higher levels of TCR activation (based on CD25 upregulation). Migratory cDC subsets were cultured with naïve OT-II T cells as in (A) (expt. = 5). (C) TA1 cognate antigens are presented by mesenteric lymph node (MLN) migratory dendritic cells. $1.5 \times 10^3$ TA1- or CT6-expressing T cell hybrids were co-cultured in vitro with $5 \times 10^4$ ex vivo MLN migratory cDCs (MHCII$^{hi}$ CD11c$^{int}$). NFAT-GFP expression in hybrids was analyzed after 40 hr (expt. = 2). (D) Expression of a Th1-inducing TCR (TA1) in CT6 cells dominantly inhibits CT6 pTreg cell differentiation in vivo. Naïve CT6 T cells were retrovirally transduced with either TA1 or a non-reactive control TCR (T7-2 or T9). Untransduced CT6, CT6 co-expressing TA1, or CT6 co-expressing T7-2 or T9 were transferred into each mouse; mice received either $5 \times 10^4$ of each TCR (closed shapes) or $2 \times 10^5$ of each TCR (open shapes). Expression of *Foxp3*$^{IRES-GFP \ or \ Thy1.1}$ and CXCR3 were quantified in the distal MLN (dMLN) after 7 days. Each dot represents the mean of indicated experiments (A and B), an individual co-culture well (C), or an individual mouse (D). Mean ± SEM shown (A–D). p-values from mixed effects analysis of repeated measures of DC subsets with Tukey's multiple comparisons test (A and B); Tukey's multiple comparisons test (C and D); *p<0.05, **p<0.01, ***p<0.001, ****p<0.0001.

The online version of this article includes the following source data and figure supplement(s) for figure 7:

**Source data 1.** Migratory conventional dendritic cell (cDC) induction of OT-II peripheral regulatory T (pTreg) cells in vitro is dependent on T cell receptor (TCR) stimulation, Figure 7A raw data.

**Source data 2.** In vitro, CD103$^+$ SP cDC1s have increased ability to induce OT-II peripheral regulatory T (pTreg) cells at higher levels of TCR activation, Figure 7B raw data.

**Figure supplement 1.** Conventional dendritic cell (cDC)-mediated peripheral regulatory T (pTreg) cell differentiation in vivo is recessive to effector T cell development.

flagellin and is likely important for Teff cell responses to pathogenic bacteria such as Salmonella (*Kinnebrew et al., 2012*; *Uematsu et al., 2008*). We therefore hypothesized that pTreg cell selection is 'recessive', requiring all T cell-interacting antigen-presenting cDC subsets to be permissive to pTreg cell selection such that activated T cells at a minimum retain the ability to upregulate FOXP3. During inflammatory conditions, a subset of cDCs may then 'dominantly' induce fully differentiated effector T cells via the production of cytokines that block FOXP3 induction, even if the T cells subsequently encounter cDCs that facilitate pTreg cell generation.

To test this hypothesis, we used adoptive transfer of modified CT2 or CT6 T cells into wild-type, Helicobacter-colonized hosts, as the blockade of in vivo pTreg cell differentiation by Th1 differentiation has not been reported to our knowledge. First, we cultured naïve Helicobacter-specific cells in the presence of Th1- or Th2-polarizing conditions in vitro to model the possibility that pTreg cell differentiation in vivo would be dominantly inhibited by the presence of Th1- or Th2-inducing cytokines during activation. After activation in Th1 or Th2 conditions in vitro, in vivo pTreg cell differentiation of CT2 T cells was completely abrogated (*Figure 7—figure supplement 1C*). To address that in vitro activation itself did not lead to an inability to undergo pTreg cell differentiation in vivo, we cultured CT2 cells in 'Th0'-inducing conditions, which permitted normal pTreg cell differentiation after adoptive transfer in vivo (*Figure 7—figure supplement 1C*). These data also show that in vitro T cell activation without the induction of Foxp3 ('Th0') does not result de facto in the generation of fully differentiated CD44$^{hi}$ effector T cells that are no longer capable of becoming pTreg cells, consistent with our previous observation that the frequency of Foxp3$^+$ cells continues to increase after 1 week in transferred transgenic T cells (*Nutsch et al., 2016*).

To address the possibility that in vitro Th1 differentiation does not mimic in vivo conditions, we expressed a Th1-directing TCR (TA1) in CT6 T cells, hypothesizing that the TA1 TCR will engage antigen in a Th1 environment not normally experienced by CT6 cells under homeostatic conditions. We first confirmed that the stimulatory antigen for TA1, which is thus far uncharacterized, is presented by DCs in the MLN using T cell hybridoma cells expressing GFP under a minimal NFAT promoter (*Ise et al., 2010*). When TA1- or CT6-expressing hybrids were co-cultured with MLN migratory DCs from Helicobacter-free mice, TA1 but not CT6-expressing hybrids were activated (*Figure 7C*). Looking then in vivo, CT6 cells that co-expressed TA1, but not a non-reactive control TCR, failed to induce Foxp3 but expressed the Th1 marker CXCR3 (*Figure 7D*). CT6 cells that co-expressed TA1 also proliferated more than cells that expressed a non-reactive control TCR, indicating that CT6 engagement on a 'tolerogenic' cDC was not sufficient to inhibit cell division (*Figure 7—figure supplement 1D*). Taken together, these data support a 'recessive' model in which pTreg cell differentiation is subdominant to a Th1 response in the same localized tissue environment.

## Discussion

The generation of gut commensal-specific pTreg cells is a critical mechanism for maintaining gut homeostasis, and yet the cDC subsets that present antigen to naïve T cells and direct T cell development remain unclear (*Mowat, 2018*). Here, we provide direct evidence identifying the cDC subsets that are capable of presenting Helicobacter antigens. Unexpectedly, we found that all migratory cDC subsets, both CD103$^+$ and CD103$^-$, contained cells capable of presenting Helicobacter antigens ex vivo. The use of genetic models with large reductions in specific cDC subsets supported the notion that CD103$^+$ cDC1s and cDC2s were neither required for antigen presentation nor pTreg cell generation in vivo. These observations argue against the notion that a singular tolerogenic cDC subsets presents gut antigens and induces pTreg cell generation (*Hasegawa and Matsumoto, 2018*; *Scott et al., 2011*; *Takenaka and Quintana, 2017*; *Tanoue et al., 2016*), and lend support to the idea that CD103$^-$ CD11b$^+$ cDC2s are important or redundant for intestinal pTreg cell induction (*Kim et al., 2018*; *Veenbergen et al., 2016*). Our data also suggest a 'recessive' model in which all gut cDC subsets are permissive to pTreg cell generation during homeostasis, but where pTreg cell selection can be 'dominantly' blocked by the existence of pro-inflammatory cDCs during inflammation or infection, even when homeostatic pTreg cell inducing cDCs remain.

One issue with the study of intestinal cDC subsets in vivo is the limitation of the genetic tools used for cDC subset depletion. For example, *Batf3*$^{-/-}$ mice have been used to deplete cDC1s, but many cDC1s remain in mucosal-associated tissues in these mice (*Tussiwand et al., 2015*). Additionally, deletion of *Batf3* in other cell types may affect immune cell homeostasis beyond loss of cDC1

cells. Newer models such as the $Xcr1^{DTR}$ mouse appears to be a more efficient alternative to $Batf3^{-/-}$ mice for depletion of cDC1s (*Yamazaki et al., 2013*). Similarly, $Irf8^{\Delta149en/\Delta149en}$ mice only lose IRF8 expression in cDC1 cells, resulting in specific loss of cDC1 development (*Durai et al., 2019*). However, there may still be some limitations. $Xcr1^{DTR}$ may not deplete immature cDC1s (5–10%) and may deplete a small fraction (5%) of CD103$^+$ CD11b$^+$ cDC2s in the MLN (*Yamazaki et al., 2013*). Also, longer term studies with diphtheria toxin are not possible due to the generation of neutralizing antibodies. Our data suggest that $Irf8^{\Delta149en/\Delta149en}$ also does not completely block cDC1 generation in the gut, specifically the colon. A direct experimental comparison of $Xcr1^{DTR}$ and $Irf8^{\Delta149en/\Delta149en}$ is therefore needed. Nonetheless, we observed that the total number of cDC1s in $Irf8^{\Delta149en/\Delta149en}$ mice was greatly reduced compared to control mice without grossly affecting Helicobacter-specific pTreg cell differentiation in vivo.

Similarly, the elimination of CD103$^+$ CD11b$^+$ cDC2s has been achieved via Itgax-Cre mediated deletion of *Irf4* or *Notch2*, as well as CD207-DTA. However, we found the effects of these deletions on Helicobacter-specific pTreg cell generation varied substantially, with *Irf4* and CD207-DTA showing little to no effect and deletion of *Notch2* in cDCs inhibiting pTreg cell generation (*Nutsch et al., 2016*). Since all of these genetic models lead to a loss of DP cDC2s, our current interpretation of these discrepant effects is that DP cDC2s are not required for pTreg cell selection and that the deletion of *Notch2* in other cDC subsets leads directly or indirectly to a defect in pTreg cell induction. However, we acknowledge that CD207-DTA depletion of DP cDC2s is not complete in the context of cDC1 depletion, and that a caveat of these and other genetic models is that it remains possible that the remaining cDCs in the targeted subset may be sufficient for pTreg cell induction. Thus, these data provide a cautionary tale for studying cDC function using these Cre-deletion models that were originally used to analyze cDC development and suggest that new genetic models are required to better understand the role of DP and CD11b$^+$ SP cDC2 subsets in vivo.

An unexpected observation from our data was that all migratory cDC subsets presented Helicobacter antigens. Research into acquisition of luminal antigens by cDCs has suggested that specific cDC subsets may be specialized for this process. Small intestinal CD103$^+$ cDC1s or cDC2s have been reported to extend trans-epithelial dendrites into the intestinal lumen to phagocytose pathogenic *Salmonella typhimurium* during infection (*Farache et al., 2013*). Goblet associated passages have been shown to deliver soluble dietary antigens to lamina propria CD103$^+$ cDC1s or cDC2s in the small intestine (*McDole et al., 2012*). CD103$^+$ SP cDC1s have also been shown to be uniquely capable of cross-presenting antigens from apoptotic intestinal epithelial cells, which could contain phagocytosed luminal antigens (*Cerovic et al., 2015*). Yet, to our knowledge, a direct assessment of the in vivo loaded endogenous commensal antigens presented by cDC subsets in the MLN has not been reported. Although our data shows that all cDC subsets acquire antigen, the loading may not be uniform, as we observed subtly different effects of cDC subset depletion on CT2 and CT6 T cell proliferation. This may be due to differences between *H. typhlonius* and *H. apodemus* colonization, or in the specific proteins recognized by CT2 and CT6. Future studies are required to determine if antigen loading on all cDC subsets is generalizable to other gut bacteria.

The lack of selectivity of Helicobacter antigens for presentation by a particular cDC subset, coupled with the in vivo observation that substantial deletion of CD103$^+$ cDC1s and cDC2s does not markedly impact pTreg cell selection to Helicobacter, favors the notion that Treg cell selection in vivo utilizes redundant cDC subsets rather than specialized 'tolerogenic' CD103$^+$ cDC1s and cDC2s. While these genetic models of cDC subset deficiency have clearly been shown to impact the immune system (*Durai et al., 2019*; *Welty et al., 2013*), we do acknowledge that the loss of CD103$^+$ cDC1s and cDC2s, while substantial, is not complete and therefore we cannot exclude the possibility that the remaining CD103$^+$ cDC1s and cDC2s are sufficient. Although we did not observe a correlation of CT2 and CT6 Foxp3 induction with CD103$^+$ cDC numbers (cDC1 or cDC2s), new tools with complete deletion of CD103$^+$ cDC1s and cDC2s will be required to definitively address the role of these cDCs in pTreg cell generation in vivo.

Our results showing that CD103$^+$ cDC1s are more efficient than cDC2s for pTreg cell induction in vitro is consistent with previous reports on the total CD103$^+$ cDC population (*Coombes et al., 2007*; *Sun et al., 2007*). It therefore remains possible that certain bacterial or food antigens in the gut may utilize this quantitative difference between the cDC subsets in pTreg cell induction in vivo. Although future studies are required, our data imply that these antigens would be present at high dose and would be preferentially acquired by cDC1s. However, the fact that cDC1-deficient mice do

not appear to suffer from spontaneous gut immunopathology (*Welty et al., 2013*) suggests that these antigens do not represent a major component of gut tolerance.

Our data suggest that commensal-specific pTreg cell differentiation can take place when cDCs carrying cognate antigen are permissive to pTreg cell induction such that they do not deliver a signal that terminally differentiates T cells into a FOXP3⁻ effector T cell lineage. This fits well with data suggesting that the MLN stroma environment is intrinsically pTreg cell-inducing (*Cording et al., 2014*; *Esterházy et al., 2019*; *Fletcher et al., 2015*; *Pezoldt et al., 2018*), which would influence the function of all cDC subsets at homeostasis in the absence of inflammatory cues. In accordance with this idea, other groups have found that the tolerogenic milieus of skin and tumors induce a pro-Treg cell-inducing gene signature that is common across migratory cDC subsets (*Anandasabapathy et al., 2014*; *Nirschl et al., 2017*). It is notable that CD103⁻ CD11b⁺ cDC2s did not express factors associated with increased cDC-mediated Treg induction such as RALDH activity or *Ido1* and *Itgb8* expression (*Boucard-Jourdin et al., 2016*; *Coombes et al., 2007*; *Matteoli et al., 2010*; *Sun et al., 2007*), which may explain the decreased ability of CD103⁻ CD11b⁺ cDC2s to promote Foxp3 induction in vitro compared to CD103⁺ cDC1s. This also implies that there may be additional cDC or environmental factors that promote Foxp3 induction in the gut.

We do not believe that every T:DC interaction during homeostasis is required to induce a binary Treg vs T effector cell outcome. Even if the T cell does not upregulate FOXP3 after activation by the cDC, the cells may adopt a 'Th0' phenotype which retains the ability to subsequently upregulate FOXP3 or other lineage-specifying transcription factors. Support for this comes from our observation that CNS1-deficient T cells, which show little to no upregulation of Foxp3 at 1 week, appear to undergo induction of Foxp3 at later times points (*Nutsch et al., 2016*). Similarly, the frequency of Foxp3⁺ cells continues to increase after 1 week of adoptive transfer of naïve T cells, although this may also be due to relative expansion (*Nutsch et al., 2016*). Finally, in vitro activated Foxp3⁻ T cells transferred in vivo can still undergo Foxp3 induction given the right TCR as long as the cells were not exposed to Th1- or Th2-inducing cytokines in vitro. Thus, pTreg cell selection may be termed 'recessive' as all cDC subsets presenting antigen would be required to be permissive for this process.

We predict that the homeostatic pTreg cell selection process can be disrupted by T cell encounter with cDCs expressing signals that induce effector phenotypes and block FOXP3 induction. As proof of concept, we showed that TCR transgenic cells exposed to Th1 or Th2 cytokines in vitro no longer undergo pTreg cell selection in vivo, confirming previous in vitro results (*Caretto et al., 2010*; *Wei et al., 2007*). In addition, TCR transgenic cells co-expressing a Th1 TCR, which engages cDCs other than that seen by the Helicobacter-specific TCR, also failed to upregulate Foxp3. Together, these data suggest a model whereby pTreg cell selection by tolerogenic cDCs is 'recessive' to cDCs that dominantly induce effector T cell development. This mechanism provides a satisfying model for operation of the gut adaptive immune system. Anti-inflammatory pTreg production would dominate during homeostatic conditions in the presence of typical commensal microbiota, whereas pro-inflammatory effector T cells, and not pTreg cells, would emerge when a pathogen is sensed. If antigen is carried by multiple cDC subsets, then naïve antigen-specific T cells could potentially differentiate into multiple effector T cell subsets in the correct contexts. The result is a careful balance between steady-state functional tolerance and focused anti-microbial responses that help preserve gut function in the context of time-varying microbial populations.

## Materials and methods

**Key resources table**

| Reagent type (species) or resource | Designation | Source or reference | Identifiers | Additional information |
|---|---|---|---|---|
| Antibody | Anti-mouse CD3ε (clone# 145–2 C11) FITC, PE, and BV421 | Biolegend | 100305/07/35 | Dilution: 1:300 |

*Continued on next page*

*Continued*

| Reagent type (species) or resource | Designation | Source or reference | Identifiers | Additional information |
|---|---|---|---|---|
| Antibody | Anti-mouse/human B220 (clone# RA3-6B2) APC/Cy7 and A700 | Biolegend | 103223/31 | Dilution: 1:750 |
| Antibody | Anti-mouse CD19 (clone# 6D5) APC/Cy7 | Biolegend | 115529 | Dilution: 1:750 |
| Antibody | Anti-mouse I-Ab (clone# AF6-120.1) APC and PerCP/Cy5.5 | Biolegend | 116417/15 | Dilution: 1:750 |
| Antibody | Anti-mouse CD11c (clone# N418) PE/Cy7 and BV605 | Biolegend | 117317/33 | Dilution: 1:750 |
| Antibody | Anti-mouse/human CD11b (clone# M1/70) BV711 | Biolegend | 101241 | Dilution: 1:750 |
| Antibody | Anti-mouse CD103 (clone# 2E7) BV421 | Biolegend | 121421 | Dilution: 1:300 |
| Antibody | Anti-mouse CD4 (clone# RM4-5) BV711, PE, and PB | Biolegend | 100549/11/34 | Dilution: 1:750 |
| Antibody | Anti-mouse CD25 (clone# PC61) APC, BV605, PerCP/Cy5.5, and PE-Cy7 | Biolegend | 102011/35/29/15 | Dilution: 1:750 |
| Antibody | Anti-mouse/human CD44 (clone# IM7) APC/Cy7 and BV605 | Biolegend | 103027/47 | Dilution: 1:750 |
| Antibody | Anti-mouse CD62L (clone# MEL-14) APC/Cy7 and BV605 | Biolegend | 104427/37 | Dilution: 1:750 |
| Antibody | Anti-mouse FOXP3 (clone# FJK-16s) FITC | Thermo Fisher | 11-5773-82 | Dilution: 1:200 |
| Antibody | Anti-mouse GATA3 (clone# 16E10A23) PE | Biolegend | 653803 | Dilution: 1:30 |
| Antibody | Anti-mouse RORγt (clone# B2D) APC | Thermo Fisher | 17-6981-80 | Dilution: 1:200 |
| Antibody | Anti-mouse TBET (clone# 4B10) PE/Cy7 | Biolegend | 644823 | Dilution: 1:200 |
| Antibody | Anti-mouse HELIOS (clone# 22F6) A647 | Biolegend | 137208 | Dilution: 1:200 |
| Antibody | Anti-mouse CXCR3 (clone# CXCR3-173) BV421 | Biolegend | 126521 | Dilution: 1:300 |
| Antibody | Anti-mouse Thy1.1 (clone# 30-H12) PE/Cy7 | Biolegend | 105325 | Dilution: 1:750 |
| Antibody | Anti-mouse LAP (clone# TW7-16B4) BV421 | Biolegend | 141407 | Dilution: 1:100 |
| Antibody | Anti-mouse CD51 (clone# RMV-7) PE | Biolegend | 104105 | Dilution: 1:200 |
| Antibody | Anti-mouse CD80 (clone# 16-10A1) APC | Biolegend | 104713 | Dilution: 1:750 |
| Antibody | Anti-mouse CD86 (clone# GL-1) BV605 | Biolegend | 105037 | Dilution: 1:300 |

*Continued on next page*

*Continued*

| Reagent type (species) or resource | Designation | Source or reference | Identifiers | Additional information |
|---|---|---|---|---|
| Antibody | Anti-mouse CD40 (clone# 3/23) APC | Biolegend | 124611 | Dilution: 1:750 |
| Antibody | Anti-mouse CD273 (clone# TY25) APC | Biolegend | 107210 | Dilution: 1:300 |
| Antibody | Anti-mouse CD274 (clone# 10F.9G2) BV421 | Biolegend | 124315 | Dilution: 1:300 |
| Antibody | Anti-mouse XCR1 (clone# ZET) APC | Biolegend | 148205 | Dilution: 1:300 |
| Antibody | Anti-mouse CD36 (clone# CRF D-2712) PE | Becton Dickinson | 562702 | Dilution: 1:200 |
| Antibody | Anti-mouse CD45.1 (clone# A20) PE, APC, and PE/Cy7 | Biolegend | 110707/14/29 | Dilution: 1:750 |
| Antibody | Anti-mouse CD45.2 (clone# 104) PE, APC, PE/Cy7, and A700 | Biolegend | 109807/14/29/21 | Dilution: 1:750 |
| Antibody | Anti-mouse VB6 (clone# RR4-7) PE and APC | Biolegend | 140003/5 | Dilution: 1:750 |
| Antibody | Anti-mouse Va2 (clone# B20.1) APC/Cy7 and PerCP/Cy5.5 | Biolegend | 127818/13 | Dilution: 1:750 |
| Antibody | Anti-mouse VB5 (clone# MR9-4) PE | Biolegend | 139503 | Dilution: 1:750 |
| Antibody | Anti-mouse F4/80 (clone# BM8) PE/Cy7 | Biolegend | 123113 | Dilution: 1:750 |
| Antibody | Anti-mouse SIRPα (clone# P84) A700 | Biolegend | 144021 | Dilution: 1:750 |
| Antibody | Anti-mouse CD24 (clone# M1/69) PE | Biolegend | 101807 | Dilution: 1:750 |
| Antibody | Anti-mouse CD101 (clone# Moushi101) PE/Cy7 | Thermo Fisher | 50-112-3316 | Dilution: 1:300 |
| Strain, strain background (*Mus musculus*) | OT-II TCR transgenic mice | The Jackson Laboratory (JAX) | #004194 | |
| Strain, strain background (*Mus musculus*) | *Ccr7*[GFP] knockin/knockout mice | The Jackson Laboratory (JAX) | #027913 | |
| Strain, strain background (*Mus musculus*) | Itgax-Cre mice | The Jackson Laboratory (JAX) | #008068 | |
| Strain, strain background (*Mus musculus*) | *Irf4*[fl/fl] mice | The Jackson Laboratory (JAX) | #009380 | |
| Strain, strain background (*Mus musculus*) | CT2 TCR transgenic mice | *Nutsch et al., 2016* | | |
| Strain, strain background (*Mus musculus*) | CT6 TCR transgenic mice | *Nutsch et al., 2016* | | |

*Continued on next page*

*Continued*

| Reagent type (species) or resource | Designation | Source or reference | Identifiers | Additional information |
|---|---|---|---|---|
| Strain, strain background (*Mus musculus*) | *Rag1*[−/−] mice | The Jackson Laboratory (JAX) | #002216 | |
| Strain, strain background (*Mus musculus*) | *Foxp3*[IRES-GFP] mice | The Jackson Laboratory (JAX) | #006772 | |
| Strain, strain background (*Mus musculus*) | *Foxp3*[IRES-Thy1.1] mice | *Liston et al., 2008* | | |
| Strain, strain background (*Mus musculus*) | *Irf8*[Δ149en/Δ149en] mice; formerly *Irf8* +32 5'[−/−] | *Durai et al., 2019* | | |
| Strain, strain background (*Mus musculus*) | *Batf3*[−/−] mice | *Hildner et al., 2008* | | |
| Strain, strain background (*Mus musculus*) | *Zbtb46*[GFP] mice | *Satpathy et al., 2012* | | |
| Strain, strain background (*Mus musculus*) | CD207-DTA mice; formerly huLangerin-DTA | *Kaplan et al., 2005* | | |

## Mice

Animal breeding and experiments were performed in a specific pathogen-free animal facility using protocols approved by the Washington University Institutional Animal Care and Use Committee (protocol #20170036). Littermates were used for all comparisons of control and cDC-deficient mice. CT2, CT6, and OT-II transgenic mice were bred to *Rag1*[−/−] and *Foxp3*[IRES-GFP] (*Lin et al., 2007*) or *Foxp3*[IRES-Thy1.1] (*Liston et al., 2008*).

## Adoptive transfer and harvest of T cells

Lymph nodes and spleens were harvested from congenially marked (Ly5.1, Ly5.2, or Ly5.1/2) CT2, CT6, or OT-II transgenic mice. Naïve T cells (CD4$^+$ Foxp3$^-$ CD25$^-$ CD62L$^{hi}$ CD44$^{lo}$) were sorted on a FACSAria IIu (Becton Dickinson) and stained with CTV (Thermo Fisher #C34571). $5 \times 10^4$ (CT2/OT-II), or $10^5$ (CT6) cells were retro-orbitally injected into 3–4-week-old mice on day 0 unless otherwise noted. For OT-II transfer in an oral tolerance model, mice were gavaged with 50 mg of OVA (Sigma #A5503) on days 1 and 2. CT2 and CT6 cells were analyzed by flow cytometry (Flowjo) in the dMLN on day 7, while OT-II cells were analyzed from the whole MLN on day 8. For analysis of polyclonal cLP T cells, colons were processed as in *Nutsch et al., 2016*. Briefly, colons were cleaned and incubated in RPMI (Thermo Fisher # SH3025502) with 3% bovine calf serum, 20 mM HEPES, dithiothreitol (Sigma #43819), and EDTA for 20 min at 37°C with constant stirring. Tissue was further digested with 28.3 µg/ml Liberase TL (Sigma #5401020001) and 200 µg/ml deoxyribonuclease I (Sigma #DN25), with continuous stirring at 37°C for 30 min. Digested tissue was forced through a Cellector tissue sieve (Bellco Glass) and passed through a 40 µm cell strainer. Transcription factors were stained using the FOXP3/Transcription Factor Staining Buffer Set (Thermo Fisher #00-5523-00).

## Preparation of DCs

DCs were harvested from the dMLN of 3–5-week-old Ly5.1 *Foxp3*[IRES-GFP] mice in our colony for ex vivo and fluorescence-activated cell sorting (FACS) experiments, and from cDC deficient mice for qPCR experiments. dMLN were dissociated in RPMI containing 5% bovine calf serum, penicillin/streptomycin, 1 mM sodium pyruvate, non-essential amino acids, 50 µM beta-mercaptoethanol, 65.8 µg/ml collagenase VIII (Sigma # C2139), and 0.2 U/ml dispase (Thermo Fisher # CB-40235) for 45 min at 37°C with continuous stirring. For culture and qPCR, DCs were blocked in 10 µg/ml anti-

CD16/CD32 (BioXCell #BE0307) and sorted using the following markers: migratory cDCs (CD3ε⁻ B220⁻ CD19⁻ IAb^hi CD11c^int), resident cDCs (CD3ε⁻ B220⁻ CD19⁻ IAb^int CD11c^hi), CD103⁺ SP (CD3ε⁻ B220⁻ CD19⁻ IAb^hi CD11c^int CD103⁺ CD11b⁻), DP (CD3ε⁻ B220⁻ CD19⁻ IAb^hi CD11c^int CD103⁺ CD11b⁺), and CD11b⁺ SP (CD3ε⁻ B220⁻ CD19⁻ IAb^hi CD11c^int CD103⁻ CD11b⁺). RALDH activity was quantified using Aldefluor (Stemcell #01700).

Ex vivo cDC-T cell simulation assay cDCs with indicated phenotypes were sorted by flow cytometry from the dMLN of 10–15 mice naturally colonized by vertical transmission of *H. typhlonius* and *H. apodemus.* $10^4$ cDCs were cultured with $2.5 \times 10^4$ naïve (CD4⁺ Foxp3⁻ CD25⁻ CD44^lo CD62L^hi) CT2, CT6, or OT-II T cells sorted from transgenic mice. Cells were cultured in 96-well U-bottom plates for 2 days at 37°C in 200 µl complete DMEM (Thermo Fisher #SH3008101) with 10% FBS, glutamax, 50 µM beta-mercaptoethanol, and 1 mM sodium pyruvate (10 mM HEPES, non-essential amino acids, and penicillin/streptomycin). *H. typhlonius* and *H. apodemus* were cultured as in *Chai et al., 2017*, autoclaved, filtered, and quantified for protein concentration by Bradford protein assay. Where noted, either 2 ng/µl of autoclaved *H. typhlonius* or 1.4 ng/µl of autoclaved *H. apodemus* was added to individual cultures. Where noted, cDC:T cell cocultures were incubated with 0.025 µg/µl αMHC Class II blocking antibody (a different clone from the non-blocking IAb antibody used to sort cDCs) (BioXCell #BE010). For cultures with OT-II T cells, indicated concentrations of OVA^323-339 were added to cultures. For T cell hybridoma stimulation by ex vivo cDCs, $10^4$ CT6 or TA1 hybridomas were incubated with $10^4$ ex vivo MLN cDCs as described above. Where noted, 20 µg/ml anti-CD3 (BioXCell BE0001-1) was added to each well as a positive control. qPCR cDCs with indicated markers were sorted into lysis buffer and processed for RNA with the Nucleospin RNA XS kit (Machery-Nagel 740902). cDNA was synthesized with SuperScript III Reverse Transcriptase (ThermoFisher 18080085). qPCR was carried out with Luminaris Color HiGreen qPCR Master Mix (ThermoFisher K0391) using a LightCycler 480 (Roche) for real-time quantitative RT-PCR. Transcripts were normalized to *Gapdh* and quantified in two technological replicates. Primer pairs used were: *Gapdh* (F: 5′-ACAAGATGG TGAAGGTCGGTGTGA-3′, R: 5′-AGCTTCCCATTCTCAGCCTTGACT-3′) (*Zhou et al., 2014*), *Tgfβ1* (F: 5′-GCTACCATGCCAACTTCTGT-3′, R:5′-CGTAGTAGACGATGGGCAGT-3′) (*Kuczma et al., 2009*), *Tgfβ2* (F: 5′-TCGACATGGATCAGTTTATGCG-3′, R: 5′-CCCTGGTACTGTTGTAGATGGA-3′) (*Li et al., 2013*), *Tgfβ3* (F: 5′-CGAGTGGCTGTTGAGGAGA-3′, R: 5′-GCTGAAAGGTGTGACATGGA-3′) (*de Verteuil et al., 2014*), *Il10* (F: 5′-AGTGGAGCAGGTGAAGAGTG-3′, R: 5′-TTCGGAGAGAGG TACAAACG-3′) (*Kuczma et al., 2009*), *Itgβ6* (F: 5′-AAACGGGAACCAATCCTCTGT-3′, R: 5′-GCTTC TCCCTGTGCTTGTAGG-3′) (*Tatler et al., 2016*), *Itgβ8* (F: 5′-CTGAAGAAATACCCCGTGGA-3′, R: 5′-ATGGGGAGGCATACAGTCT-3′) (*Melton et al., 2010*), *Ido1*(F: 5′-GAAGGATCCTTGAAGAC-CAC-3′, R: 5′-GAAGCTGCGATTTCCACCAA-3′) (*Orabona et al., 2006*).

## 16S rRNA sequencing

Fecal DNA was purified via column (Zymogenetics) and used in triplicate PCR of the bacterial V4 hypervariable region of 16S rRNA using barcoded primers described previously (*Caporaso et al., 2011*). PCR products were sequenced using the Illumina MiSeq platform ($2 \times 250$ bp paired end reads), and ASVs and taxonomy including species designations if possible (silva 1.32) determined by dada2 (*Callahan et al., 2016*). ASV data were analyzed by phyloseq (v1.32), vegan (v2.4), and DESeq2 (v.1.28) in R (3.6).

## In vitro polarization and transduction of T cells

24-well plates were coated with 10 µg/ml anti-CD3ε (BioXCell BE0001-1) overnight at 4°C. Wells were washed with PBS and plated with $5–15 \times 10^5$ naïve CT6 T cells in D10 and 1 µg/ml anti-CD28 (BioXCell BE0015-1) in Th0 conditions: 10 µg/ml anti-TGFβ (BioXCell BE0057), 5 µg/ml anti-IL12 (BioXCell BE0052), 5 µg/ml anti-IFNγ (BioXCell BE0054), and (5 mg/ml anti-IL4 (BioXCell BE0045)); Th1-polarizing conditions: 10 ng/ml IL-12 (Peprotech #210–12); or Th2-polarizing conditions: 10 ng/ml IL-4 (Peprotech #214–14). For in vitro polarization experiments, naïve CT2 cells were plated on anti-CD3ε and cultured for 48 hr in Th0 or Th1 conditions, after which $1.5 \times 10^5$ cells were injected into each host. For TCR transduction, TCRα chains utilized were TA1: TRAV14-3*01 CDR3 AASETGNTG-KLI; T7-2: TRAV7-4*01 CDR3 AASEHWSNYQLI; and T9-1: TRAV9-1*02 CDR3 AVSAPNTNKVV. TCRα chain retroviral transduction was performed as described in *Hsieh et al., 2004* using TransIT-293 (Thermo Fisher # MIR2700). Note that the transduced cells all share the same original TCli TCRβ

chain (*Hsieh et al., 2004*). CT6 cells were plated on anti-CD3 in Th0 media, transduced 27 hr later, and left to rest in Th0 media for 48–66 hr. Foxp3⁻ cells were then sorted for TA1-transduced (TdTomato⁺), control TCR-transduced (Thy1.1⁺), or untransduced cells. $5 \times 10^4$ (co-injection of transduced TCRs totaling $1.5 \times 10^5$ T cells per mouse) or $2 \times 10^5$ (injection of single transduced TCR plus untransduced control cells) of each TCR were injected into each host. T cell hybridomas expressing GFP under a minimal NFAT promoter (*Ise et al., 2010*) were retrovirally transduced with CT6 or TA1 TCRα chains as previously described (*Chai et al., 2017*; *Lathrop et al., 2011*).

## Statistical analysis

Graphpad Prism v7 was used for statistical and graphical analysis unless noted. CTV division index was obtained by manually defining CTV peaks for each cell population and using the formula $\frac{\sum_0^i \left( i \times \frac{N_i}{2^i} \right)}{\sum_0^i \left( \frac{N_i}{2^i} \right)}$, where (i) = the maximum number of cell divisions in the population and (*N*) = the number of cells in a given CTV peak (i). Student's t-test was used for between-subject analyses with two groups. One-way ANOVA with Tukey's multiple comparisons tests was used for between-subjects analysis with greater than two groups and one independent variable. Two-way ANOVA with Sidak's multiple comparisons tests was used for between-subjects analysis with greater than two groups and two independent variables. Where stated, the mixed effects model with Tukey's multiple comparisons tests were used instead of two-way ANOVA for repeated measures data with missing values. For in vivo experiments, each dot represents data from an individual host. Bars indicate mean ± SEM. *p<0.05, **p<0.01, ***p<0.001, ****p<0.0001.

## Acknowledgements

We thank Nicole Santacruz and Patricia Hsieh for technical assistance (Wash.U.). We thank Jessica Hoisington-Lopez at the Center for Genome Sciences and Systems Biology for Miseq sequencing expertise (Wash.U.).

## Additional information

### Funding

| Funder | Grant reference number | Author |
| --- | --- | --- |
| National Institutes of Health | R01 AI079187 | Chyi-Song Hsieh |
| National Institutes of Health | R01 AI136515 | Chyi-Song Hsieh |
| Burroughs Wellcome Fund | | Chyi-Song Hsieh |
| National Institutes of Health | F30 DK111071 | Emilie V Russler-Germain |
| National Institutes of Health | T32 GM007200 | Emilie V Russler-Germain |

The funders had no role in study design, data collection and interpretation, or the decision to submit the work for publication.

### Author contributions

Emilie V Russler-Germain, Conceptualization, Data curation, Formal analysis, Funding acquisition, Validation, Investigation, Visualization, Methodology, Writing - original draft, Project administration, Writing - review and editing; Jaeu Yi, Data curation, Formal analysis, Writing - review and editing; Shannon Young, Data curation, Formal analysis, Validation, Investigation, Writing - review and editing; Katherine Nutsch, Conceptualization, Data curation, Formal analysis, Investigation, Methodology, Writing - review and editing; Harikesh S Wong, Data curation, Software, Formal analysis, Investigation, Visualization, Methodology, Writing - review and editing; Teresa L Ai, Investigation, Methodology, Writing - review and editing; Jiani N Chai, Resources, Methodology, Writing - review and editing; Vivek Durai, Daniel H Kaplan, Resources, Writing - review and editing; Ronald N Germain, Resources, Software, Supervision, Visualization, Methodology, Project administration, Writing - review and editing; Kenneth M Murphy, Resources, Supervision, Writing - review and editing; Chyi-

Song Hsieh, Conceptualization, Resources, Software, Formal analysis, Supervision, Funding acquisition, Visualization, Methodology, Writing - original draft, Project administration, Writing - review and editing

### Author ORCIDs
Emilie V Russler-Germain [ID] https://orcid.org/0000-0002-2036-6697
Jiani N Chai [ID] http://orcid.org/0000-0003-2560-8218
Ronald N Germain [ID] http://orcid.org/0000-0003-1495-9143

### Ethics
Animal experimentation: All animal experiments were performed in strict accordance with the guidelines of the Institutional Animal Care and Use Committee at Washington University (Protocol Number: 20170036).

### Decision letter and Author response
Decision letter https://doi.org/10.7554/eLife.54792.sa1
Author response https://doi.org/10.7554/eLife.54792.sa2

## Additional files
### Supplementary files
• Transparent reporting form

### Data availability
16S rRNA sequencing data has been deposited in ENA (PRJEB42640).

The following dataset was generated:

| Author(s) | Year | Dataset title | Dataset URL | Database and Identifier |
|---|---|---|---|---|
| Russler-Germain EV, Hsieh CS | 2021 | Fecal 16S samples from CD207-DTA and Irf8-Delta149en cDC deficient mice | https://www.ebi.ac.uk/ena/browser/view/PRJEB42640 | European Nucleotide Archive, PRJEB42640 |

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
