## [Decision Letter]

**Acceptance summary:**

We appreciated the investigation of the nature of the dendritic cell subset involved in regulating immune responses to *Helicobacter* spp in mouse colonies showing that under normal conditions, a variety of DC subsets may drive the generation of peripheral regulatory T cells and confirming that this is not necessarily a specific property of CD103^+^ DC. Under inflammatory conditions, the development of Treg may be over-ridden by effector T cell differentiation, despite the presence of tolerogenic DC.

**Decision letter after peer review:**

Thank you for submitting your article "Gut *Helicobacter* presentation by multiple dendritic cell subsets enables context-specific regulatory T cell generation" for consideration by *eLife*. Your article has been reviewed by three peer reviewers, one of whom is a member of our Board of Reviewing Editors, and the evaluation has been overseen by Wendy Garrett as the Senior Editor. The reviewers have opted to remain anonymous.

The reviewers have discussed the reviews with one another and we have drafted this decision to help you prepare a revised submission.

Summary:

In this paper, Russler-Germain et al. explore the nature of the dendritic cell subset involved in regulating immune responses to *Helicobacter* spp in the mouse colon. By examining the anatomically appropriate draining lymph node, the authors conclude that under normal conditions, a variety of DC subsets may drive the generation of regulatory T cells and that this is not a specific property of CD103^+^ DC as has been proposed previously. Indeed, CD103-CD11b^+^ DC may be the most effective subset in this respect. Under inflammatory conditions, the usual development of Treg may be over-ridden, despite the presence of tolerogenic DC. Herein, the authors propose an interesting hypothesis in which tolerance is a "recessive" phenomenon. The work uses a variety of animal models to dissect individual DC subsets in selective ways and analyses the resulting findings with caution, which is commendable. Although many of the results are negative, they are important, as they challenge several previous assumptions about tolerance to intestinal antigens that were often made based on less specific models or on use of anatomically less precise tissues. However, the work also supports an earlier finding that there may be flexibility among DC subsets in their ability to drive Treg cells against protein antigens in different segments of the intestine, with CD103-CD11b^+^ DC appearing to be particularly important in the colon. There are also a number of useful cautionary points raised about models used to assess DC subset function, generation of Treg and tolerance in this manuscript.

Essential revisions:

We appreciated the conclusion that no single DC subset can account for tolerance in the intestine given the paradigm of CD103^+^ "tolerogenic" DC in other papers. We also appreciated the fact that the authors had attempted to improve on previous approaches for depleting individual DC subsets in vivo.

1) Whilst we felt the overall message was important and would be a useful addition to the field there are concerns that the current work cannot exclude possible roles for contaminating cells from other phenotypic subsets and that in a number of places the results are over-interpreted. The major issue is therefore not with the data as such, but with the authors' strong counter-dogmatic interpretations that likely remain open to dispute. The reviewers agree that the work would be acceptable if presented with very clear caveats rather than "dogma challenging." A number of previous studies have also provided evidence that there is redundancy in the ability of different DC subsets to induce Treg differentiation. These previous studies have obvious caveats (explored in the Introduction), but the current study does not entirely eliminate such issues. We therefore ask that the interpretations, discussion and conclusions need to be more nuanced throughout.

2) The authors are obviously fully aware of the heterogeneity amongst the population referred to as "CD103^+^ DC" and indeed, address this issue experimentally. However it would be appropriate for this to be highlighted from the earliest stage of the manuscript, particularly when referring to the earlier in vivo and in vitro studies that did not discriminate between the CD103^+^CD11b^+^ and CD103^+^CD11b- subsets present amongst this population. Ideally this would mean defining throughout which subsets of CD103^+^ DC had been used, employing the cDC1 and cDC2 nomenclature that has become standard.

While the evidence that CD103-CD11b^+^ DC play a more significant role than anticipated is fairly persuasive, some of the data used to support this idea are somewhat over-interpreted. For instance, these cells actually express very few of the molecules that are associated with "tolerogenic" DC in the intestine, such as b8 integrin, RALDH, IDO. This should be acknowledged, as should the findings that confirm previous work showing that cDC1 are the population expressing some of these molecules such as b8 and RALDH.

3) Both CD103^+^CD11b- and CD103^+^CD11b^+^ subsets have been proposed to contribute to pTreg generation, so it is not necessarily surprising that the models used by the authors to deplete one or the other of these subsets ( Irf8 +32 5'^-/-^, hLangDTA, Irf4ΔDC, Notch2ΔDC) do not affect the frequency of Helicobacter-specific pTreg cell generation. Particularly since the loss of one CD103^+^ population is often compensated for by an increase in the other. The key experiment therefore becomes the Irf8 +32 5'^-/-^ hLangDTA cross, which depletes both CD103^+^ subsets. However, even in this model ~20% of the remaining DCs are CD103^+^ (total numbers reduced from 30,000 to 10,000) so it is difficult to see how this data supports the authors conclusion that "Contrary to the prevailing notion suggesting that CD103^+^ DCs are important, loss of these DCs in vivo using genetic models did not affect the frequency of Helicobacter-specific pTreg cell generation." Were the remaining CD103^+^ SP and DP DCs analysed in these mice to determine if they could still looked like their WT counterparts (as in Figure 6) or could still efficiently induce Foxp3 expression in vitro? I feel that the conclusions of the paper should be softened to account for this issue.

4) While clearly a more effective system to target both the cDC1 and cDC2 subsets of CD103^+^ DC, it has to be acknowledged that some of both subset remain in the hLangDTA Irf8 +32 5'^-/-^ mice.

5) The extent of *Helicobacter* colonisation in the mice and composition of the microbiota needs to be reported. One aspect which is not mentioned is whether any of the various deletional models might be influenced by the microbiome which is likely to be distinct in different laboratories using the same strains. In this respect, do the authors have any information on the composition of the microbiota in their different strains used here?

6) The immunohistology analysis was thought to be fraught with potential confounding issues and should be removed.

7) Whilst the authors acknowledge that 15-25% of CD103-CD11b^+^ cells are likely to be macrophages, this needs to be more clearly emphasised and discussed.

[Editors' note: further revisions were suggested prior to acceptance, as described below.]

Thank you for submitting your article "Gut *Helicobacter* presentation by multiple dendritic cell subsets enables context-specific regulatory T cell generation" for consideration by *eLife*. Your article has been reviewed by two reviewers, and the evaluation has been overseen by a Reviewing Editor and Wendy Garrett as the Senior Editor. The reviewers have opted to remain anonymous.

The reviewers have discussed the reviews with one another and the Reviewing Editor has drafted this decision to help you prepare a revised submission.

We would like to draw your attention to changes in our revision policy that we have made in response to COVID-19 (https://elifesciences.org/articles/57162). Specifically, we are asking editors to accept without delay manuscripts, like yours, that they judge can stand as *eLife* papers without additional data, even if they feel that they would make the manuscript stronger. Thus the revisions requested below only address clarity over claims and caveats in interpretation.

The manuscript provides potentially important data on the nature of the dendritic cell which is responsible for maintaining immunological tolerance against bacteria in the intestine. The data support the view that each of the several subsets of dendritic cells can generate regulatory T cells under different circumstances, indicating the flexibility of the dendritic cell population and the importance of having redundant mechanisms to maintain tolerance in this tissue.

Summary:

The authors are thanked for their replies to the original comments and for the additional experimental data they have provided. Overall, the manuscript was considered much improved and the conclusions that intestinal DC subsets may have overlapping abilities to induce regulatory T cells are substantiated better.

Revisions:

We would be glad if the authors could address the following issues mainly in the text to provide appropriate caveats of the interpretations of their data and acknowledgement of the contributions already available in the literature. There is also one additional point of negative controls which should be provided.

No negative controls are shown for T cells stimulated without DC or antigen are provided for the assays shown in Figure 2. This is important, as some of the proportions of CD25^+^ T cells stimulated by the different DC subsets are low.

Other points to deal with caveats, interpretations and acknowledgments of other contributions.

1) It is probably fair to say that the authors have still not acknowledged fully that the idea of redundancy between intestinal DC functions is not as novel as they suggest. Furthermore, they do not state that the study by Veenbergen et al. cited here showed directly that CD103-CD11b^+^ DC could induce tolerance to OVA in the colon, a finding that is replicated here for Helicobacter.

2) In respect of this subset, it is also still not acknowledged that these cells do not express significant levels of Aldh, Itgb8 and Ido, the mediators that have been most strongly associated with tolerogenic functions of DC.

3) Although the authors now show data indicating potential contamination of the CD103-CD11b^+^ "DC" subset with macrophages, this finding is ignored subsequently in the manuscript. While this may not contradict the conclusion that no single subset of genuine DC can account for the generation of Treg, it does complicate interpretation of the studies of gene expression by sorted subsets. In particular, several of the genes interpreted as being potentially related to tolerance induction by CD103-CD11b^+^ "DC" are characteristic of intestinal macrophages, including IL10, av integrin and TGFb.

4) While the findings presented here are highly important in showing that none of the conditional deletion strategies is fully effective in eliminating the relevant subset of DC, this does remain a critical issue in interpreting the current data and has not yet been acknowledged adequately.

5) The lack of control data from the straight Notch-2 δ DC mice remains an important caveat when interpreting the experiments in the double KO mice.

6) Were all the mice used at such young ages? If so, this difference between the current and previous studies warrants more detailed acknowledgement than here.

7) It is important to note that the studies of Treg induction by Coombes et al., 2007 and Sun et al., 2007 did not show a role for cDC1, as these experiments were performed using total CD103^+^ DC and before it was appreciated that this population contained both cDC1 and cDC2.

---

## [Author Response]

Essential revisions:We appreciated the conclusion that no single DC subset can account for tolerance in the intestine given the paradigm of CD103^+^ "tolerogenic" DC in other papers. We also appreciated the fact that the authors had attempted to improve on previous approaches for depleting individual DC subsets in vivo.1) Whilst we felt the overall message was important and would be a useful addition to the field there are concerns that the current work cannot exclude possible roles for contaminating cells from other phenotypic subsets and that in a number of places the results are over-interpreted. The major issue is therefore not with the data as such, but with the authors' strong counter-dogmatic interpretations that likely remain open to dispute. The reviewers agree that the work would be acceptable if presented with very clear caveats rather than "dogma challenging." A number of previous studies have also provided evidence that there is redundancy in the ability of different DC subsets to induce Treg differentiation. These previous studies have obvious caveats (explored in the Introduction), but the current study does not entirely eliminate such issues. We therefore ask that the interpretations, discussion and conclusions need to be more nuanced throughout.

We have amended the manuscript to better reflect the nuances of the data. These changes include modifications to language in the Abstract, Results, and Discussion. We have also added this limitation to the Discussion.

2) The authors are obviously fully aware of the heterogeneity amongst the population referred to as "CD103^+^ DC" and indeed, address this issue experimentally. However it would be appropriate for this to be highlighted from the earliest stage of the manuscript, particularly when referring to the earlier in vivo and in vitro studies that did not discriminate between the CD103^+^CD11b^+^ and CD103^+^CD11b- subsets present amongst this population. Ideally this would mean defining throughout which subsets of CD103^+^ DC had been used, employing the cDC1 and cDC2 nomenclature that has become standard.

We have amended the manuscript to include CD103^+^ subset descriptions and cDC1/cDC2 nomenclature where appropriate.

While the evidence that CD103-CD11b^+^ DC play a more significant role than anticipated is fairly persuasive, some of the data used to support this idea are somewhat over-interpreted. For instance, these cells actually express very few of the molecules that are associated with "tolerogenic" DC in the intestine, such as b8 integrin, RALDH, IDO. This should be acknowledged, as should the findings that confirm previous work showing that cDC1 are the population expressing some of these molecules such as b8 and RALDH.

As suggested, we have added additional discussion of these previous findings in the Results.

3) Both CD103^+^CD11b- and CD103^+^CD11b^+^ subsets have been proposed to contribute to pTreg generation, so it is not necessarily surprising that the models used by the authors to deplete one or the other of these subsets ( Irf8 +32 5'^-/-^, hLangDTA, Irf4ΔDC, Notch2ΔDC) do not affect the frequency of Helicobacter-specific pTreg cell generation. Particularly since the loss of one CD103^+^ population is often compensated for by an increase in the other. The key experiment therefore becomes the Irf8 +32 5'^-/-^ hLangDTA cross, which depletes both CD103^+^ subsets. However, even in this model ~20% of the remaining DCs are CD103^+^ (total numbers reduced from 30,000 to 10,000) so it is difficult to see how this data supports the authors conclusion that "Contrary to the prevailing notion suggesting that CD103^+^ DCs are important, loss of these DCs in vivo using genetic models did not affect the frequency of Helicobacter-specific pTreg cell generation." Were the remaining CD103^+^ SP and DP DCs analysed in these mice to determine if they could still looked like their WT counterparts (as in Figure 6) or could still efficiently induce Foxp3 expression in vitro? I feel that the conclusions of the paper should be softened to account for this issue.

We previously did not perform in vitro functional experiments using the remaining cDCs as their numbers were low. Therefore, we have softened the conclusions of the manuscript to account for this issue as discussed in response to point #1 above. As suggested, we performed analysis of TCR stimulation and tolerogenic markers in remaining CD103^+^ cDC1s and cDC2s in CD207-DTA::*Irf8*^D149en/D149en^ mice. We found that the only marker that was significantly changed was CD86 expression, which was reduced in both subsets of cDCs in CD207-DTA::*Irf8*^D149en/D149en^ mice (Figure 6—figure supplement 2B).

4) While clearly a more effective system to target both the cDC1 and cDC2 subsets of CD103^+^ DC, it has to be acknowledged that some of both subset remain in the hLangDTA Irf8 +32 5'^-/-^ mice.

We have noted this issue more clearly in the text.

5) The extent of Helicobacter colonisation in the mice and composition of the microbiota needs to be reported. One aspect which is not mentioned is whether any of the various deletional models might be influenced by the microbiome which is likely to be distinct in different laboratories using the same strains. In this respect, do the authors have any information on the composition of the microbiota in their different strains used here?

As suggested, we analyzed bacterial microbiomes by 16S rRNA sequencing of colonic contents. We did not observe obvious differences in Shannon diversity or β diversity by Unifrac or Bray-Curtis NMDS. Frequencies of *Helicobacter typhlonius* and *Helicobacter apodemus* were not markedly impacted (Figure 6—figure supplement 1).

6) The immunohistology analysis was thought to be fraught with potential confounding issues and should be removed.

Although we believe that the immunohistochemistry showing no obvious physical associations of CT2 and CT6 with any migratory cDC subset in vivo was informative, we have removed these data and analysis from the manuscript as suggested.

7) Whilst the authors acknowledge that 15-25% of CD103-CD11b^+^ cells are likely to be macrophages, this needs to be more clearly emphasised and discussed.

We have expanded our discussion of this topic in the Results section. In brief, we state that our previous in vivo data using *Zbtb46^DTR^* mice showing the importance of Zbtb46^+^ cells for in vivo CT2 and CT6 activation and pTreg cell generation (Nutsch et al., 2016) suggest that the Zbtb46^GFP–^ cells in our sorted CD103^–^CD11b^+^ APCs would not likely be a major contributor to in vitro responses.

[Editors' note: further revisions were suggested prior to acceptance, as described below.]

Revisions:We would be glad if the authors could address the following issues mainly in the text to provide appropriate caveats of the interpretations of their data and acknowledgement of the contributions already available in the literature. There is also one additional point of negative controls which should be provided.No negative controls are shown for T cells stimulated without DC or antigen are provided for the assays shown in Figure 2. This is important, as some of the proportions of CD25^+^ T cells stimulated by the different DC subsets are low.

We routinely performed T cell only (no cDC) controls for the in vitro experiments and include these re-analysis resulting in changed Figure 1, Figure 2, Figure 1—figure supplement 1B, and Figure 7—figure supplement 1A,B. Data interpretations remained unchanged.

Other points to deal with caveats, interpretations and acknowledgments of other contributions.1) It is probably fair to say that the authors have still not acknowledged fully that the idea of redundancy between intestinal DC functions is not as novel as they suggest. Furthermore, they do not state that the study by Veenbergen et al. cited here showed directly that CD103-CD11b^+^ DC could induce tolerance to OVA in the colon, a finding that is replicated here for Helicobacter.

We have edited our Introduction to more clearly discuss the data presented by Veenbergen et al. (Introduction, Results and Discussion). Note that this study does not appear to show equivalent in vivo Foxp3 induction in *Batf3*^–/–^ vs WT mice, but only in vitro Foxp3 induction in the presence of TGFb. Equivalent functional tolerance in *Batf3^–/–^* vs WT mice in vivo was reported.

2) In respect of this subset, it is also still not acknowledged that these cells do not express significant levels of Aldh, Itgb8 and Ido, the mediators that have been most strongly associated with tolerogenic functions of DC.

We have acknowledged these findings with our interpretation in the Discussion.

3) Although the authors now show data indicating potential contamination of the CD103-CD11b^+^ "DC" subset with macrophages, this finding is ignored subsequently in the manuscript. While this may not contradict the conclusion that no single subset of genuine DC can account for the generation of Treg, it does complicate interpretation of the studies of gene expression by sorted subsets. In particular, several of the genes interpreted as being potentially related to tolerance induction by CD103-CD11b^+^ "DC" are characteristic of intestinal macrophages, including IL10, av integrin and TGFb.

We have addressed this caveat in the text.

4) While the findings presented here are highly important in showing that none of the conditional deletion strategies is fully effective in eliminating the relevant subset of DC, this does remain a critical issue in interpreting the current data and has not yet been acknowledged adequately.

We have clarified this caveat in the Results and Discussion.

5) The lack of control data from the straight Notch-2 δ DC mice remains an important caveat when interpreting the experiments in the double KO mice.

We acknowledge that it would have been useful to also include experiments with *Notch2^fl/fl^* controls from the same litters. Unfortunately, we do not have this breeding set up to address this concern. We believe that the manuscript does not depend on the *Notch2*-DDC data, so we have removed this data (Figure 4—figure supplement 2A–C were removed, with Figure 4—figure supplement 2D moved to Figure 4—figure supplement 1E).

6) Were all the mice used at such young ages? If so, this difference between the current and previous studies warrants more detailed acknowledgement than here.

We have added justification for our use of young host mice.

7) It is important to note that the studies of Treg induction by Coombes et al., 2007 and Sun et al., 2007 did not show a role for cDC1, as these experiments were performed using total CD103^+^ DC and before it was appreciated that this population contained both cDC1 and cDC2.

We have clarified this important point in the Introduction.